# Experimental and Theoretical Study on the Synergistic Inhibition Effect of Pyridine Derivatives and Sulfur-Containing Compounds on the Corrosion of Carbon Steel in CO_2_-Saturated 3.5 wt.% NaCl Solution

**DOI:** 10.3390/molecules23123270

**Published:** 2018-12-11

**Authors:** Junlei Tang, Yuxin Hu, Zhongzhi Han, Hu Wang, Yuanqiang Zhu, Yuan Wang, Zhen Nie, Yingying Wang

**Affiliations:** 1School of Chemistry and Chemical Engineering, Southwest Petroleum University, Chengdu 610500, China; tangjunlei@126.com (J.T.); hyxin519@163.com (Y.H.); zhuline518@163.com (Y.Z.); wangyuan0508@126.com (Y.W.); yingyingwanglyon@126.com (Y.W.); 2CNPC Engineering Technology Research Company Limited., No. 40 Jintang Road, Tanggu Binhai New Area, Tianjin 300451, China; hanzhongzhi0304@163.com; 3School of Material Science and Engineering, Southwest Petroleum University, Chengdu 610500, China; 4Research Institute of Petroleum Exploration and Development, CNPC, Beijing 100083, China; niezhen@petrochina.com.cn

**Keywords:** corrosion inhibitor, synergistic effect, electrochemical measurements, theoretical calculation

## Abstract

The corrosion inhibition performance of pyridine derivatives (4-methylpyridine and its quaternary ammonium salts) and sulfur-containing compounds (thiourea and mercaptoethanol) with different molar ratios on carbon steel in CO_2_-saturated 3.5 wt.% NaCl solution was investigated by weight loss, potentiodynamic polarization, electrochemical impedance spectroscopy, and scanning electron microscopy. The synergistic corrosion inhibition mechanism of mixed inhibitors was elucidated by the theoretical calculation and simulation. The molecules of pyridine derivative compounds with a larger volume has priority to adsorb on the metal surface, while the molecules of sulfur-containing compounds with a smaller volume fill in vacancies. A dense adsorption film would be formed when 4-PQ and sulfur-containing compounds are added at a proper mole ratio.

## 1. Introduction

Carbon dioxide (CO_2_) corrosion is a serious problem in oil and gas wells and pipelines, causing unacceptable electrochemical corrosion of carbon steel, especially in oil and gas production and transmission [1,2,3,4]. Due to the low-cost and convenience, the injection of inhibitors is an economical and effective method for corrosion protection [5,6].

In general, corrosion inhibitors for carbon steel used in the oilfield industry are organic compounds that contain unsaturated bonds or atoms, such as N, O, S, etc. They can form coordination bonds with empty orbits of metal elements, and adsorb on the metal surface to protect metal material [7,8,9]. Thereinto, imidazole, quinoline, pyridine, and their derivatives are most commonly used to control corrosion owing to the structure of the nitrogen-containing heterocycle in the oilfield industry. Several researchers have discovered that these compounds adsorb parallelly on the iron-based metal surface by the action of the nitrogen-containing heterocycle to block the corrosion medium [10,11]. However, imidazoline and its derivatives have drawbacks, like their relative instability in storage, emulsifying tendency in produced water, and relative economical expense [12,13]. Regarding quinoline and its derivatives, their good water solubility enhances their application, however, researchers are uneasy to study them due to the theoretical calculations for complex molecular structures [14,15,16]. The performance of pyridine and its derivatives is similar to quinoline. Pyridine derivative molecules contain an N atom and heterocyclic group, and the simple and effective structure has been proven to have a desirable inhibition effect in strong acid solutions [17,18,19]. Researchers have probed the special structure of those compounds to bridge between the experimental results and quantum chemical theoretical studies. However, in the theoretical calculations [10,20,21], it was found that in the adsorption model of single pyridine derivatives molecules, there were uncovered voids on the metal surface.

In reality, the corrosion inhibitor of a single compound is always insufficient to protect carbon steel tubing and casing for oil production because of the severe corrosive environment. The development of inhibitors’ mixture with a synergistic inhibition effect is the economical and effective way for increasing the inhibition efficiency, decreasing the amount of usage and diversifying the application of inhibitors in aggressive medium [22]. At present, most studies have been based on experimental results to presume the synergistic mechanism of mixed corrosion inhibitors. Many researchers have noticed the synergistic effects among specific substances, between organic and inorganic compounds [23,24,25], surfactants [26,27], or two organics [28,29]. However, the synergistic mechanism of organic compounds is complex, which highly depends on the compound structure. It is worth mentioning that small sulfur-containing compounds with straight chains, such as thiourea and mercaptoethanol, have greatly different molecular structures compared with pyridine derivatives. Their protective effect for carbon steel is determined by the S atom in molecular structures [30,31,32], but the dosage cost limits its further application. They may make up the uncovered portion with pyridine derivative molecules on the metal surface. Economical and effective corrosion inhibitor mixtures can be designed if there is a significant synergistic inhibition effect between pyridine derivatives and sulfur-containing compounds, which has not been investigated to the best of our knowledge. 

More importantly, the classical adsorption model of mixed corrosion inhibitors has a synergistic effect, considering the adsorption mechanism was that the inhibitor molecule with strong bonding preferentially adsorbed on the metal surface by the way of erect or tilted arrangement, and another inhibitor was superimposed on the void to form a coherent hydrophobic film as a barrier for migration of the corrosive medium [33,34]. However, there is no sufficient and convincing reason to prove it, and the real adsorption configuration at the metal/solution interface is very difficult to be confirmed by the in-situ microscopic characterization at the molecular level. Theoretical methods, including quantum chemical calculation and molecular dynamic simulation, have proven to be the most appropriate methods for elucidating the synergistic inhibition mechanism of organic inhibitors, and many researchers have used molecular simulation methods to solve the issue of adsorption configuration. However, a large number of reported simulations only used the vacuum environment and many of them only simulated the adsorption behavior of a single molecule. The molecular dynamic simulations with respect to the actual conditions (in aqueous solution, molecular concentration, different ratios of the two molecules) have been reported very rarely [13,35,36]. However, the number of molecules in the unit volume, H_2_O molecules, and the interaction between different corrosion inhibitor molecules have important influences on the adsorption behavior. Therefore, the molecular dynamics simulation of the corrosion inhibitors’ mixture considering the H_2_O molecule, corrosion inhibitor composition, and concentration should be addressed. The reasonable adsorption model is proposed to explain the synergistic mechanism of mixed corrosion inhibitors by combining theoretical calculations with experimental results, and there are some reasons to doubt the traditional adsorption model of synergistic mechanism. 

In this paper, a derivative of pyridine, 4-methylpyridine (4-MP), and its quaternary ammonium salts (4-PQ), and sulfur-containing compounds, including thiourea (TU) and mercaptoethanol (TZ), were selected to deduce the synergistic effect in CO_2_-saturated 3.5 wt.% NaCl solution by the weight-loss method, open circuit potential (OCP), electrochemical impedance spectroscopy (EIS), potentiodynamic polarization, and scanning electron microscopy (SEM). The synergistic mechanism was proposed based on the electrochemical results analysis, quantum chemical parameters calculation, and molecular dynamics (MD) simulation.

## 2. Results and Discussion

### 2.1. Weight Loss Measurements

The corrosion inhibition effects of 4-MP, 4-PQ, TU, TZ, and their mixture with different molar ratios at 2 × 10^−4^ mol/L for Q235 steel in CO_2_-saturated 3.5 wt.% NaCl solution were studied by the weight loss test at 60 °C for 72 h immersion, as shown in Table 1 and Figure 1. Table 1 shows that 2 × 10^−4^ mol/L 4-MP or 2 × 10^−4^ mol/L 4-PQ has a poor inhibition effect, while single TU or TZ can reduce the corrosion rate at the same concentration, and the inhibition efficiency reaches 77.5% and 77.6%, respectively. There is no distinct difference between the inhibition efficiency of TU and TZ. Figure 1, indicates that the synergistic effects of 4-PQ with sulfur-containing compounds are better than that of 4-MP. The IE% of the compounded inhibitors are much higher than the sum of the IE% of individual 4-PQ and TZ or TU. The maximum values of IE% reach 89.3% and 93.1% when the molar ratio of 4-PQ and TZ or TU are all 3:1. For the synergistic system of 4-PQ and sulfur-containing compounds, the value of IE% is reduced when the molar ratio of sulfur-containing compounds increased.

### 2.2. Open Circuit Potential Monitoring

The open circuit potential measurement was performed at the beginning to ensure potentiodynamic polarization and impedance measurements under the stable system, and the results are shown in Figure 2. Figure 2a shows the results of OCP without and with single 4-PQ and TZ, and different molar ratios of 4-PQ and TZ immersed in CO_2_-saturated 3.5 wt.% NaCl solutions for 2 h. Figure 2b shows the results for the composite inhibitors of 4-PQ and TU. In inhibitor-free solution, due to the dissolution of the metal, the potential is always more negative than the initial immersion time (*t* = 0 s). In the presence of different molar ratios of inhibitors, the OCP shifts to a more positive value, because inhibitors adsorb on the metal surface. After about 1500 s, the potential reaches a steady value, and it is higher than the lack of inhibitor, indicating that the metal corrosion is protected by inhibitors [37]. Comparing to the single 4-PQ and single TU or TZ, the steady-state values of OCP with a large amount of TU or TZ are close to the single TU or TZ curve. This progressive positive of the steady state potential with the proportion of 4-PQ increasing is close to the single 4-PQ curve when the molar ratios of 4-PQ and TU or TZ are 1:1 and 5:1. However, under a certain ratio, the potential can reach the most positive value. These results indicate that the existence of a synergistic effect between 4-PQ and TU or TZ, and the molar ratio of 3:1 has the best effect of 4-PQ and TU or TZ.

### 2.3. Potentiodynamic Polarization

To further assess the synergistic effects between 4-PQ and sulfur-containing compounds, potentiodynamic polarization experiments were conducted for Q235 steel without and with 2 × 10^−4^ mol/L inhibitors at different molar ratios in CO_2_-staturated 3.5 wt.% NaCl solutions at 60 °C. Figure 3 and Figure 4 show the potentiodynamic polarization curves of blank (without corrosion inhibitors) and different corrosion inhibitors, respectively. The fitting results of the polarization curves used Tafel extrapolation by the software, C-view, in which other electrochemical parameters and the inhibition efficiencies of inhibitors can be determined from the corrosion current densities (*I*_corr_), as shown in Table 2 and Table 3, respectively.

As shown in Figure 3, compared to the blank solution, the corrosion potential (*E*_corr_) values of Q235 steel move towards positive slightly, and the anodic reaction is inhibited with the added 4-PQ and TZ at a molar ratio of 1:5, 1:3, 1:1, 3:1, and 5:1, indicating that inhibitors adsorb on the metal surface to retard anodic dissolution of mild steel and suppress cathodic reduction simultaneously. Therefore, the composite inhibitors of 4-PQ and TZ are mixed-type inhibitors, which preferentially suppress the anodic process of corrosion. Table 2 indicates the corrosion current density (*I*_corr_) values of 1:1, 3:1, and 5:1 have a more remarkable decline than the blank and individual inhibitor. Especially, the highest inhibition efficiency reaches 79.2% at the molar ratio of 3:1. However, when the molar ratio of 4-PQ and TZ is 1:3 or 1:5, the inhibition efficiency is better than adding 4-PQ alone, but lower than adding TZ alone. Therefore, 4-PQ needs to add a certain amount to promote the synergistic effect in the composite inhibitors.

From Figure 4 and Table 3, the polarization curve results of 4-PQ and TU are similar to the mixture of 4-PQ and TZ, but the synergistic inhibition efficiency is better than 4-PQ and TZ. Various cathodic curves overlap with the blank, however, the shape of the anodic curves changes greatly. The anode curve reflects the action of inhibitors. These can be divided into three regions, the slopes rapidly increase in the first region (<100 mv vs. OCP), and this process shows that the inhibitors adsorb on the metal surface and hinders the metal dissolution reaction of the anode. The second region (100~150 mv vs. OCP) has a slow growth potential and reflects the dynamic adsorption process of inhibitors on the metal surface. The third area (>150 mv vs. OCP) coincides with the blank, indicating that the inhibitor completely separates from the metal surface. The highest inhibition efficiency is 86.5% in the presence of 3:1 at 2 × 10^−4^ mol/L, and the inhibition efficiency of single 4-PQ and TU is only 16.2% and 49.4%, respectively. Thus, it can be seen that adding sulfur-containing compounds has a good synergistic effect with 4-PQ for Q235 in CO_2_-saturated 3.5 wt.% NaCl solution. 

The synergistic corrosion inhibition effect of composite inhibitors with different molar ratios was in the following order: 3:1 > 5:1 > 1:1 > 1:5 > 1:3 (4-PQ: TU or TZ). Although the values of IE% are different by different test methods, the results obtained by the polarization test show good agreement with the weight loss method.

### 2.4. Electrochemical Impedance Spectroscopy Measurements

The effect of the mixture of 4-PQ and sulfur-containing compounds on the Nyquist and Bode plots for mild steel in CO_2_-saturated 3.5 wt.% NaCl solution at 60 °C are presented in Figure 5. As seen from Nyquist plots, the working electrode immersed in different mole ratios all have a depressed semicircle with a center under the real axis at a high frequency range. This is generally attributed to the roughness of the electrode surface, the uneven distribution of the carbon steel surface, or the adsorption of the inhibitors on the surface of carbon steel [38,39]. In the whole frequency range, the appearance of capacitive semicircles, as observed in all Nyquist spectra, is attributed to the charge-transfer reaction and double layer capacitance. As labeled in Figure 5a, the diameter of 1:3 is less than 1:5 for the mixture of 4-PQ and TZ, but both of them are smaller than the diameter of TZ alone, which means the inhibition efficiency will be dominated by TZ and the synergistic effect with 4-PQ will be weak if TZ hold the main proportion. Then, the diameter increased significantly with the proportion of 4-PQ that increased. As seen in Figure 5b, the curve of 4-PQ mixed with TU at different molar ratios is the same as that mixed with TZ, but the semicircles’ diameter is larger than the mixture of 4-PQ and TZ. Notably, the diameter of the semicircle is the largest when the molar ratio of 4-PQ and TZ (or TU) is 3:1.

The absolute value of the impedance increases with the addition of inhibitors compared with the blank solution in the Bode plot (Figure 5c,d) at low frequency. Simultaneously, phase angle plots were observed to follow a similar trend as the Nyquist and Bode plots. Electrochemical impedance data were fitted with Zsimpwin software using non-linear least square fit techniques. Figure 6 shows an equivalent circuit diagram of the blank solution (Figure 6a) and an equivalent circuit diagram of two time constants used to fit data well for the solution containing mixed corrosion inhibitors (Figure 6b). A pure double-layer capacitor is frequently replaced by constant phase elements (CPE) representing leaky or non-ideal capacitors, with a view to compensating for non-homogeneity in the system to give a more accurate fit. Here, *R*_s_ stands for the solution resistance, *R*_ct_ is the charge transfer resistance, *R*_f_ represents the resistance of the protective inhibitor film formed on the Q235 surface, and *CPE*_dl_ and *CPE*_f_ represent the constant phase element of the double layer and the constant phase element of the adsorption film, respectively [25,40,41]. The extracted parameters from EIS are summarized in Table 4 and Table 5. The values of *C*_dl_ and *C*_f_ are calculated from *Y*_0_ and n as follows [42,43]:(1)$$C=Y_{0}{\left(2\pi f_{\max}\right)}^{n-1}$$
where *Y*_0_ represents the magnitude of the *CPE*, *f*_max_ is the frequency at the maximum value of the imaginary component of the impedance spectrum, and the value of n represents the deviation from the ideal behavior and it lies between 0 and 1.

The values of *R*_f_ and *R*_ct_ increased with the addition of composite inhibitors compared to the blank solution. Further, the value of *R*_t_ (*R*_f_ + *R*_ct_) reached maximum when the content of 4-PQ reached the proportion of 3:1 with sulfur-containing compounds. On the other hand, the minimum value of *C*_dl_ indicates that the thickness of the double layer is increased to the maximum or the dielectric constant is reduced to the minimum due to the adsorption of the inhibitor [43]. The mixture of 4-PQ and TU has better performance and inhibition efficiency reaches 82.8% at 3:1. This implies the corrosion inhibitors formed complete and dense film by the two substances. All of the above results reveal that the synergistic effect of 4-PQ with sulfur-containing compounds depends on the molar proportion of composite inhibitors.

### 2.5. Synergistic Mechanism

In order to further study the synergistic mechanism of mixed corrosion inhibitors, the linear polarization resistance method was used. Figure 7 shows the OCP and polarization resistance analysis (calculated from polarization curves vs. time) with and without 4-PQ and TU in different addition sequences. As can be seen from Figure 7a, curve 1 and 4 are the blank solution with the addition of mixed corrosion inhibitors at *t* = 0 s for comparison, respectively. Curve 2 is the addition of single TU at *t* = 0 s, and the values of OCP are close to curve 1 and increased slowly, then the addition of another corrosion inhibitor 4-PQ at *t* = 2400 s causes the OCP curve to rise sharply. Curve 3 shows addition in the reverse order, and is thus near to curve 4 at the beginning. However, all curves expect the blank solution (curve 1) are stabilized at the close-set positions and all the solutions contained 2 × 10^−4^ mol/L mixed inhibitors of 4-PQ and TU with 3:1 in the end. 

The resistance curves of direct current polarizations measured for 4-PQ and TU of 3:1 with different addition sequences are shown in Figure 7b and the resistances were obtained using linear fitting by the software, C-view (Appendix A). As shown in Figure 7b, the value of polarization resistance (*R*_p_) of the corrosion inhibitors mixture added at *t* = 0 min remained constant (785 Ω∙cm^2^) during the test, and is close to *R*_t_ from the impedance test results [44]. In the first 40 min of adding the single corrosion inhibitor, the *R*_p_ is lower than the mixed corrosion inhibitors. The *R*_p_ of adding single 4-PQ that makes up three quarters of 2 × 10^−4^ mol/L is 458 Ω∙cm^2^, and then TU is added at the 40th min, after which the curve of *R*_p_ shifted steeply immediately. Ultimately, the steady value is consistent with the curve of addition together at the beginning. On the other hand, the *R*_p_ of a quarter of 2 × 10^−4^ mol/L single TU is stable at 280 Ω∙cm^2^, and then the addition of another inhibitor, 4-PQ, and the *R*_p_ increases a bit and is lower than the previous situation. It reveals the concentration of TU is too low to form the dense inhibitor film on the metal surface after preferential addition, and further formed the crystalline compound with addition of 4-PQ, which accumulated above the film of TU [45], and 4-PQ cannot fill the active site to form a dense inhibitor film, resulting in the resistance being much lower than the mixtures. All of the above results indicate the adsorption order has an effect on the synergistic effect of composite inhibitors. 

### 2.6. Scanning Electron Microscopy Characterization

Figure 8 presents photographs of the corrosion morphologies of Q235 steel immersion in CO_2_-saturated 3.5 wt.% NaCl solution for 72 h at 60 °C in the absence and presence of the optimum molar ratio of mixed inhibitors. Figure 8a,b are blank solutions and added single 4-PQ in solution, respectively, and Figure 8c,d are the mixture of 4-PQ and sulfur-containing compounds with a molar ratio of 3:1. It can be seen that homogeneous corroded areas are observed on the sample surface in the blank solution (without corrosion inhibitors), the same as the added single 4-PQ. The surface of the sample immersed in solution with mixed inhibitors is smooth and uniform, and the corrosion degree is slowed down significantly. It illustrates that mixed inhibitors of 4-PQ and sulfur-containing compounds with the molar ratio of 3:1 formed a compact protective film on the surface of Q235 steel, inhibiting the corrosion of carbon steel in this medium.

### 2.7. Quantum Chemical Calculation

Quantum chemical calculation is a useful method to explain the theory of the adsorption of corrosion inhibitor molecules on the metal surface and the interaction between molecules. The parameters from quantum chemical calculations can be used to describe the global reaction activity of inhibitor molecules, including the highest occupied molecular orbital (HOMO) and lowest unoccupied molecular orbital (LUMO). It is easier to give electrons when the value of *E*_HOMO_ is increasing. For another, a lower *E*_LUMO_ indicates the energy is decreasing to accept electrons easily [46].

The frontier molecule orbital density distributions of the pyridine derivatives compounds (4-MP) and its quaternary ammonium salt (4-PQ) and two kinds of sulfur-containing compounds (TU and TZ) are shown in Figure 9. It can be seen that the distribution of the frontline orbit is mainly concentrated in the heterocyclic ring containing nitrogen and the sulfur atom. The quantum chemical calculation parameters of these inhibitors are shown in Table 6. The electronegativity (*χ*) is related to the chemical potential, where a higher value means better inhibitive performance. On the other hand, the lower absolute chemical hardness (*η*) implies higher polarizability and better inhibition efficiency. The softness is the reciprocal of *η*, thus a high value is related to greater efficiency, and the most imperative parameter is the energy difference (Δ*E*) between *E*_HOMO_ and *E*_LUMO_, and a decrease in Δ*E* leads to the increase in the activity of the reactive [47]. For four inhibitors, the Δ*E* of TU is the smallest and the next is 4-PQ, indicating that the combination of 4-PQ and TU is better. It coincides with the actual experimental results.

### 2.8. Molecular Dynamics Simulation

Molecular dynamics simulations of the adsorption process of corrosion inhibitors’ mixtures on Fe (1 1 0) surface under different conditions are presented in Figure 10. In a number of experimental studies, the adsorption of organic inhibitors relies on the lone pair electrons to form covalent bonds with the orbitals of metal atoms. N 1s exists in the following two forms: One is the Fe-N adsorption bond (399.6 ± 0.1 eV) [48,49], which is mainly contributed by the pyridine, quinolone, and the ammonium groups of quaternary ammonium salt, and another is the bonding between Fe and C-N-H_2_ (400.9 ± 0.3 eV) from thiourea [50,51]. The XPS spectrum around B.E. 162–164 eV, which is often observed on the steel surface when use S-containing corrosion inhibitors, can be ascribed to the S-Fe adsorption bond [28,52,53]. The MD simulation further proves the adsorption ability and configurations of these inhibitors at different molar ratios on metal surfaces (Appendix A). The side views of 4-PQ and TU reveals that the total number is 12 with the ratio of 1:5, 1:3, 1:1, 3:1, and 5:1, respectively. Some corrosion inhibitors are free in aqueous solution while the others are adsorbed on the Fe (1 1 0) surface at different molar ratios. The nitrogen-containing heterocyclic of 4-PQ provides the lone pairs of electrons to form covalent bonds with the empty orbitals of iron, and adsorbs flat on the active sites of the metal surface. The N and S atoms in the TU structure also play the same role.

It can be seen from the coverage degree of the iron-based surface that the corrosion inhibitors’ mixture with a molar ratio of 3:1 can achieve the maximum coverage and 1:3 is the minimum coverage. In addition, the adsorption stability of corrosion inhibitor molecules is studied by the adsorption energy of the corrosion inhibitor molecules. The higher absolute value of the adsorption energy implies the adsorption of the inhibitors is more stable on the metal surface, the inhibition effect is better, and these inhibitor molecules have the stronger synergy. Adsorption energy (*E*_adsoption_) is calculated as follows [54,55]:(2)$$E_{\mathrm{adsoption}}{\;=\;E}_{\mathrm{total}}\;-\;\left(E_{\mathrm{surface}+\mathrm{solution}}{+\;E}_{\mathrm{inhibitors}+\mathrm{solution}}\right)\;{-\;E}_{\mathrm{solution}}$$
where *E*_total_ is the total energy of the simulation system and includes the iron crystal together with the adsorbed inhibitor molecules on the iron surface and the solution; *E*_surface+solution_ stands for the total energy of the system without the inhibitors; *E*_inhibitors+solution_ is the total energy of the system without the iron crystal; and *E*_solution_ is designated as the total energy of the solution.

The results of 4-PQ and TU with different molar ratios by the calculations are presented in Table 7. The order of absolute value of *E*_adsoption_ is 3:1 > 5:1 > 1:1 > 1:5 > 1:3, which indicates that the compounding molar ratio has a great influence on the synergistic effect of corrosion inhibitors’ mixture. There is an optimal ratio between the TU and 4-PQ molecules, the parallel adsorption ability become stronger and the inhibitors film is denser on the Fe (1 1 0) surface when the number of 4-PQ increases, bringing about the best synergistic effect. However, it is seen from the side view of 5:1, the adsorption points of TU are seized by 4-PQ, of which the number continued to rise. The number of adsorbed molecules decreases on the iron surface and voids appear easily to cause the low adsorption energy, which is why the corrosion inhibition effect deteriorates. The synergistic mechanism of the compound inhibitors is that the 4-PQ with a larger molecular structure and the small molecule TU occupy adsorption sites completely to form a dense adsorption film at the optimal ratio. This result is mutually verified with the experimental results. It is also noticed that these adsorbed molecules are not regularly arranged at the metal/water interface, which is quite different from the classic adsorption model.

To quantitatively describe the protective effect of inhibitors’ film at different molar ratios, the concentration profile of water molecules was calculated in the absence and presence of corrosion inhibitors. As can be seen from Figure 11, the first concentration profile peak value of the water molecules’ curve appears at a height of 2.6 Å from the iron surface. Obviously, under the molar ratio of 1:3, the water density distribution is the largest, followed by 1:5 and 1:1. This is due to a large number of TU molecules occupying active sites, but the small molecule structure cannot fill the metal surface entirely and water molecules enter spacings at these molar ratios. At a molar ratio of 3:1, the peak of the water molecule concentration profile is the smallest value and has a narrower peak shape than other ratios, with rising from the iron surface distance. It shows that 4-PQ and TU have a high coverage at a molar ratio of 3:1 on the iron surface, which hinders the adsorption of water molecules and plays a good protective effect. Rather, the increasing number of large molecules that are adsorbed on the metal surface creates the voids under the molar ratio of 5:1, so that the small molecule cannot be filled and the water molecules’ concentration increases. The concentration profile of water molecules in the region of 3.8~4.8 Å from the Fe (1 1 0) surface appears as a trough with the addition of inhibitors at different molar ratios, especially when the concentration value of water molecules is close to zero at the molar ratio of 3:1. It indicates that the mixed inhibitors adsorbs on the Fe (1 1 0) surface and removes water molecules away from the metal surface. With the increase of the height from the surface, the difference of reduction for the concentration profile of the water molecules indicates the mixed inhibitors act to impede the water molecules adsorbed on the metal surface within a range of distance.

## 3. Materials and Methods

### 3.1. Materials

The molecular structures of 4-MP, 4-PQ, TU, and TZ and the synthesis method are shown in Figure 12. 4-PQ was synthesized by chemical reaction between 4-MP and benzyl chloride at 110 °C for 4 h.

The test specimen used in this study was Q235 steel with the following composition (wt.%): C 0.16, Si 0.30, Mn 0.53, P 0.015, S 0.004, and Fe balance. The corrosion medium was 3.5 wt.% NaCl solution, which was prepared using analytical grade NaCl and deionized water.

### 3.2. Weight Loss Measurement

The carbon steel sheet of 40 mm × 13 mm × 2 mm was abraded with a range of sandpaper (grade 280, 500, 1000) gradually, washed with deionized water, acetone, and anhydrous ethanol, and then dried with a cold drier. After weighing by an electronic analytical balance with a sensitivity of ±0.1 mg, the sample was then put into a 100 mL corrosion solution containing different inhibitors, and was deoxidized with N_2_ for 0.5 h and then saturated with carbon dioxide gas at atmospheric pressure by bubbling carbon dioxide for 0.5 h at 60 ± 1 °C for 72 h. After taking out the sample, it was immersed in the acid solution with hexamethylenetetramine for a short time to remove the corrosion product and protect from over etching, and was then cleaned, dried, and re-weighed accurately. After each weight loss experiment, the coupons were dried and weighed using an analytical balance (precision ±0.1 mg), and the mean weight loss value and the corresponding standard deviation were reported. 

The corrosion rate (*CR*) and inhibition efficiency (*IE*) were calculated from the following equation:(3)$$\mathrm{CR}=\frac{\Delta W}{\mathrm{St}}$$
(4)$$\mathrm{IE}=\frac{{\mathrm{CR}}_{0}{-\mathrm{CR}}_{\mathrm{inh}}}{{\mathrm{CR}}_{0}}\;\times \;100\%$$
where Δ*W* is the average weight loss of three coupons, *S* is the sample area, and *t* is the immersion time. *CR*_0_ and *CR*_inh_ are the corrosion rate with and without inhibitors, respectively.

### 3.3. Electrochemical Measurement Experiments

All the electrochemical measurements were carried out on a *Corr Test* electrochemical workstation (Wuhan Corrtest Instruments Corp., Ltd., Wuhan, China), which received and output experimental data by the connecting computer software, CorrTest (Ver 4.5). In this study, a conventional three electrodes test was carried out at 60 °C. The working electrode (WE) was a cylindrical electrode wrapped in Teflon through machine processing, and the exposed test area was 0.5 cm^2^. The saturated calomel electrode (SCE) was used as a reference electrode (RE) through the connecting salt bridge, and a platinum plate with a certain area was used as a counter electrode (CE). Before each experiment, the work electrode was polished by sandpaper (grade 500, 1000, 2000), and dried with a cold drier after rinsing with deionized water and alcohol. The three electrodes system was placed in the glass cell containing 100 mL of test solution pumped with N_2_ for 0.5 h to remove oxygen, then continuously bubbled with a steady flow of CO_2_ throughout the experiment with and without corrosion inhibitors in the solution. 

Firstly, the open circuit potential was measured to detect the change of the potential during immersion for 2 h. Electrochemical impedance spectroscopy measurements were performed after the open circuit potential over the frequency range from 0.01 Hz to 100 kHz at stable OCP with a 10 mV AC amplitude. Then, potentiodynamic polarization curves were conducted over a potential range from −400 mV to +400 mV (vs. OCP) at a scan rate of 0.5 mV s^−1^. The corrosion inhibition efficiency ($\eta_{P}$ and $\eta_{Z}$) was calculated from potentiodynamic polarization and EIS by means of the following equations, respectively:(5)$$\eta_{P}\;=\;\frac{I_{\mathrm{corr}(0)}\;{-\;I}_{\mathrm{corr}(\mathrm{inh})}}{I_{\mathrm{corr}(0)}}\;\times \;100\%$$
(6)$$\eta_{Z}\;=\;\frac{R_{t(\mathrm{inh})}{\;-\;R}_{t(0)}}{R_{t(\mathrm{inh})}}\;\times \;100\%$$
where *I*_corr(0)_ and *R*_t(0)_ represents the corrosion current density and total resistance in the absence of inhibitors, *I*_corr(inh)_ and *R*_t(inh)_ stands for the corrosion current density and total resistance of different inhibitor systems, respectively.

To study the order of adsorption in the synergistic effect, the liner polarization resistance tests were performed for different sequences of adding the one inhibitor at intervals of 40 min under a potential range from −10 mV to +10 mV (vs. OCP) at a sweep rate of 0.2 mV s^−1^ at 10 min intervals.

The solution and working electrode were changed after each sweep. Three to four measurements were performed for each experimental condition to estimate the repeatability, and a relative standard deviation was recorded.

### 3.4. Surface Analysis

The carbon steel sheets of 40 mm × 13 mm × 2 mm were immersed in CO_2_-saturated 3.5 wt.% NaCl solution without and with 2 × 10^−4^ mol/L mixed inhibitors of 4-PQ and TU (or TZ), of which the molar ratio was 3:1 at 60 °C for 72 h, washed by deionized water, dried with a cold drier, and then tested by SEM (EVO MA15, ZEISS).

### 3.5. Computational Details

The M06-2x hybrid functional method was used to optimize the geometric structure of the inhibitor in combination with a 6-311+G (d, p) basis set and the quantum chemical parameters were calculated through the Gaussian 09 software package. The relevant quantum chemical parameters, including the energies of LUMO (*E*_LUMO_) and HOMO (*E*_HOMO_), energy gap (Δ*E*), absolute chemical hardness (*η*), electrochemical potential (*μ*), electronegativity (*χ*), the number of electron transfer (Δ*N*), and electrophilicity index (*ω*), for the four inhibitors were considered [19].

Molecular mechanics simulation was performed in the Forcite module by Materials Studio 8.1 software. The Geometry optimized function was used to optimize the geometry of five combinations containing 4-PQ and TU at different molar ratios in the water—Fe (1 1 0) surface system under the force field of COMPASS. Fe (1 1 0) is a density packed surface and has the most stabilization, so we chose the Fe (1 1 0) surface to simulate the adsorption process more reasonably [56]. The smart optimization algorithm was selected and the included convergence standard was set to 2 × 10^−5^ kcal/mol and the maximum number of optimized steps was 20,000 steps. The molecular dynamics (MD) simulation and the dynamics function was used to study the interaction between five combinations and the Fe (1 1 0) surface in a simulation box (27.3 Å × 27.3 Å × 120.5 Å) with periodic boundary conditions to model a representative part of the interface devoid of any arbitrary boundary effects. The MD simulation was performed at 333 K under canonical ensemble (NVT). The initial velocity of each particle was generated by the Maxwell-Boltzmann distribution, and the Newton’s equation of motion was calculated by the Velocity Verlet algorithm. The van der Waals interaction and electrostatic effect were calculated by the Atom Based method and the Ewald method under 15 Å cut-off distance, respectively.

## 4. Conclusions

(1) In the CO_2_-saturated 3.5 wt.% NaCl solution, the inhibition effect of single 4-MP or 4-PQ was poor, but the corrosion inhibition effect was improved remarkably by using the mixed corrosion inhibitors with different molar ratios at 2 × 10^−4^ mol/L. It indicates there is a significant synergistic inhibition effect between 4-PQ and sulfur-containing compounds (TZ or TU) in such corrosive conditions. Furthermore, the mixed corrosion inhibitors mainly influence the anode process of carbon steel corrosion.

(2) The corrosion inhibition performance of the mixed corrosion inhibitors depends on the molar ratio between two mixed compounds in the following order: 3:1 > 5:1 > 1:1 > 1:5 > 1:3 (4-PQ: TZ or TU). At the molar ratio of 1:3 or 1:5, the corrosion inhibition behavior of the mixed corrosion inhibitors was similar to the single TZ or TU, because the large number of sulfur-containing compounds decreased the chance of 4-PQ molecules accessing the metal surface. The synergistic corrosion inhibition effect between 4-PQ and sulfur-containing compounds increased with the increase of the content of 4-PQ in the mixed corrosion inhibitors, but the corrosion inhibition effect decreased after the molar ratio reached 3:1.

(3) The mechanism of synergistic effects of the mixed corrosion inhibitors was revealed by electrochemical measurements, theoretical calculation, and MD simulation. The large-volume pyridine derivatives compound molecules have a priority to adsorb on the metal surface, while the small sulfur-containing compounds molecules fill in vacancies. The two molecules were in plane adsorption on the metal surface, and a dense protective film can be formed when two compounds are at the proper molar ratio, performing good corrosion resistance.

## Figures and Tables

**Figure 1 molecules-23-03270-f001:**
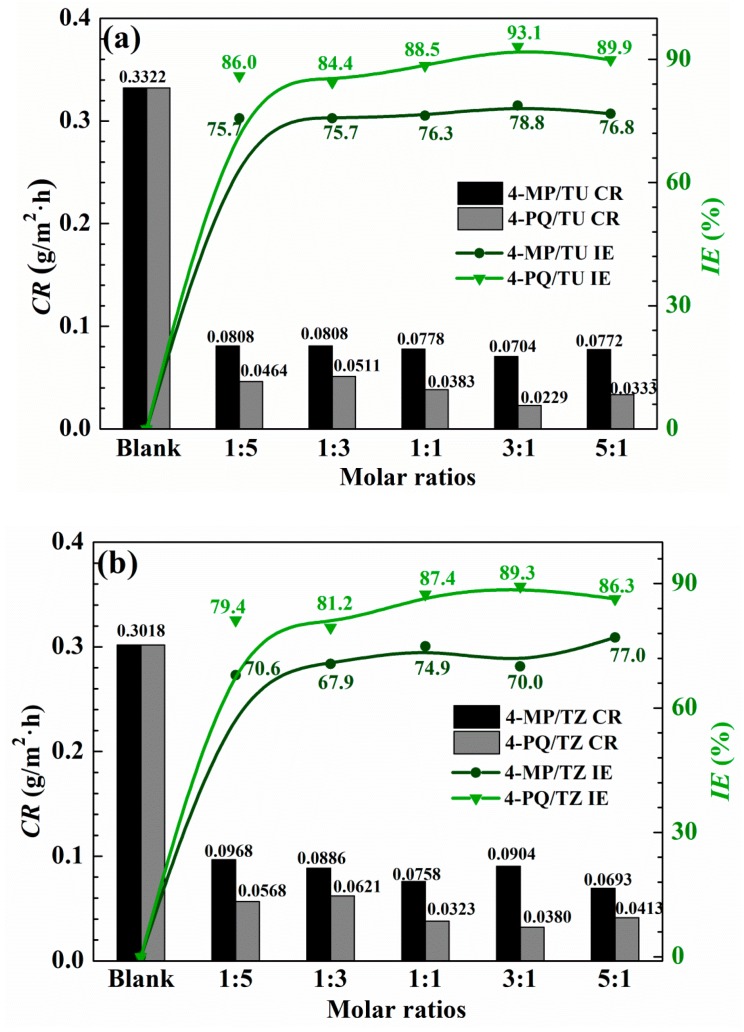
Corrosion rate and inhibition efficiency obtained from weight loss measurement of Q235 steel with different mole ratios of (**a**) 4-MP (4-PQ)/TU and (**b**) 4-MP (4-PQ)/TZ in CO_2_-saturated 3.5 wt.% NaCl solution at 60 °C.

**Figure 2 molecules-23-03270-f002:**
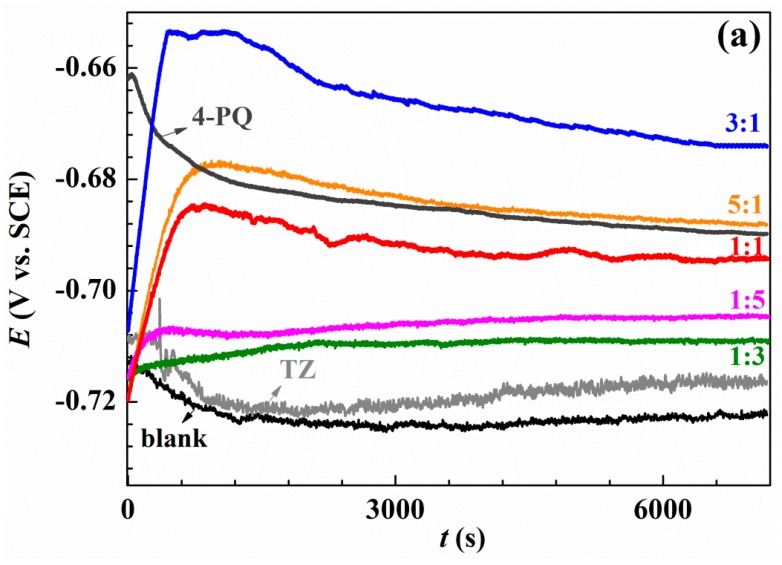
Variation of the open circuit potential with immersion time recorded of Q235 steel in CO_2_-saturated 3.5 wt.% NaCl solution at 60 °C without and with (**a**) a single inhibitor 4-PQ or TZ, and different molar ratios of 4-PQ and TZ; (**b**) single inhibitor 4-PQ or TU, and different molar ratios of 4-PQ and TU.

**Figure 3 molecules-23-03270-f003:**
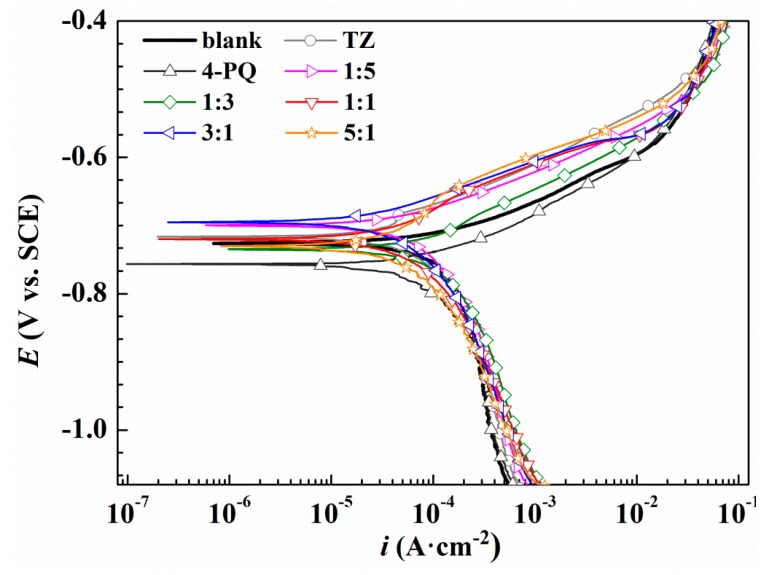
Polarization curves of Q235 steel in CO_2_-saturated 3.5 wt.% NaCl solution with or without different molar ratios of 4-PQ and TZ or individual inhibitor at 60 °C.

**Figure 4 molecules-23-03270-f004:**
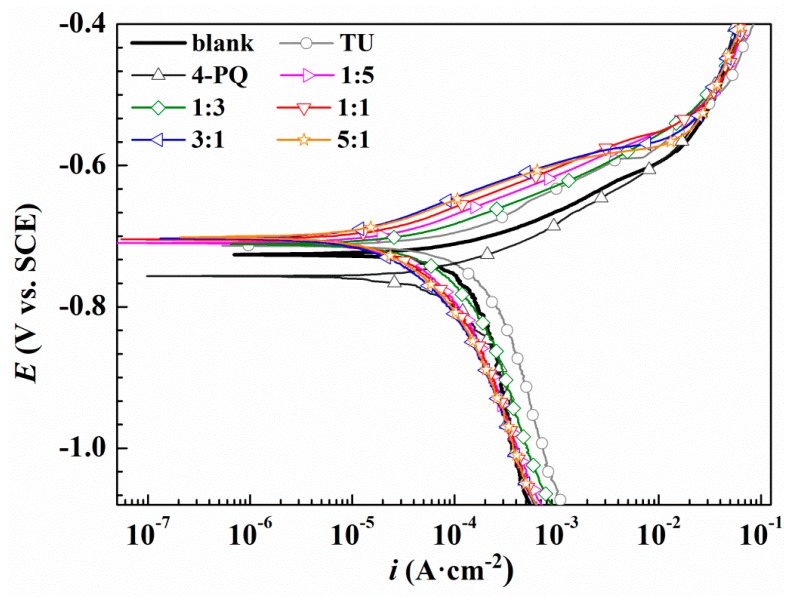
Polarization curves of Q235 steel in CO_2_-saturated 3.5 wt.% NaCl solution with or without different molar ratios of 4-PQ and TU or individual inhibitor at 60 °C.

**Figure 5 molecules-23-03270-f005:**
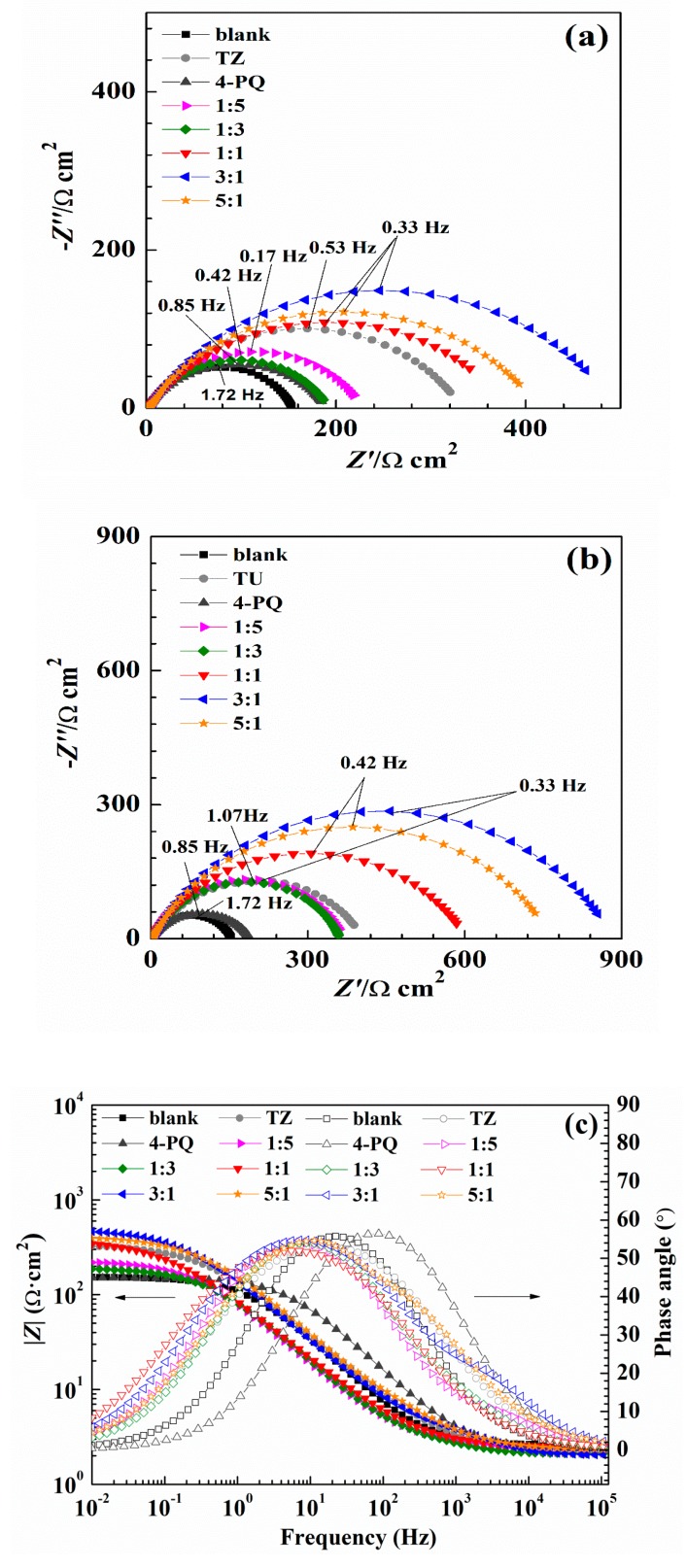
Nyquist (**a**–**d**) Bode plots of Q235 steel in CO_2_-saturated 3.5 wt.% NaCl solution with or without different mole ratios of 4-PQ and sulfur-containing compounds or individual inhibitor at 60 °C.

**Figure 6 molecules-23-03270-f006:**
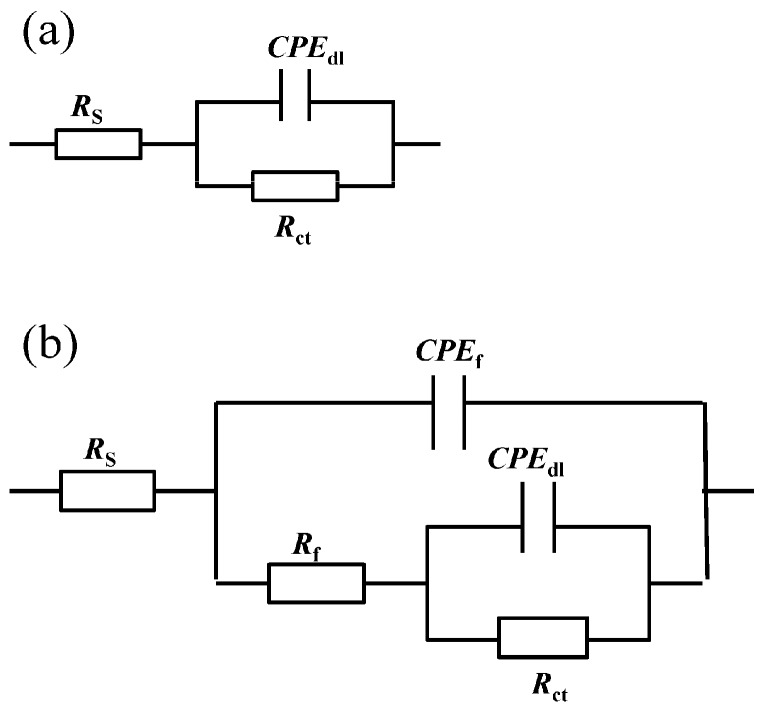
The equivalent circuit used to model the electrochemical behavior of Q235 steel in CO_2_-saturated 3.5 wt.% NaCl solution containing (**a**) a blank and (**b**) individual and mixture of 4-PQ and sulfur-containing compounds.

**Figure 7 molecules-23-03270-f007:**
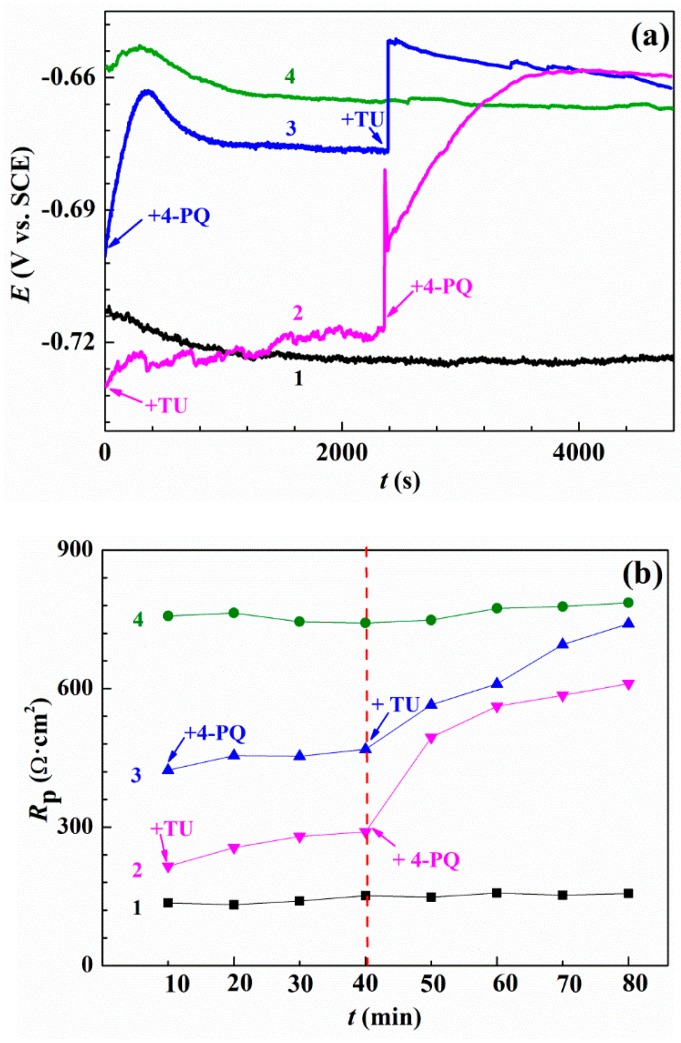
OCP measurements and polarization resistance analysis: (**a**) Variation of the open circuit potential and (**b**) values of polarization resistance without or with 4-PQ and TU in different addition sequences. Curve 1 and 4 in all the graphs stands for blank and adding corrosion inhibitors mixture contained 4-PQ and TU at the beginning, respectively. Curve 2 represents adding TU at the beginning, then adding 4-PQ at the time of the 40th min. Curve 3 represents adding 4-PQ at the beginning, then adding TU at the time of the 40th min.

**Figure 8 molecules-23-03270-f008:**
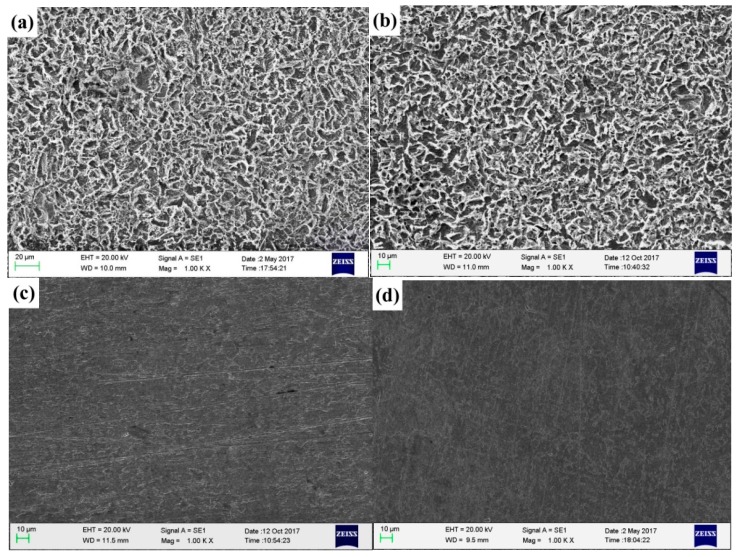
The corrosion morphology of Q235 steel obtained after immersion in CO_2_-saturated solution for 72 h: (**a**) Without inhibitors, (**b**) with 2 × 10^−4^ mol/L 4-PQ, (**c**) with 2 × 10^−4^ mol/L 4-PQ and TZ at 3:1, and (**d**) with 2 × 10^−4^ mol/L 4-PQ and TU at 3:1.

**Figure 9 molecules-23-03270-f009:**
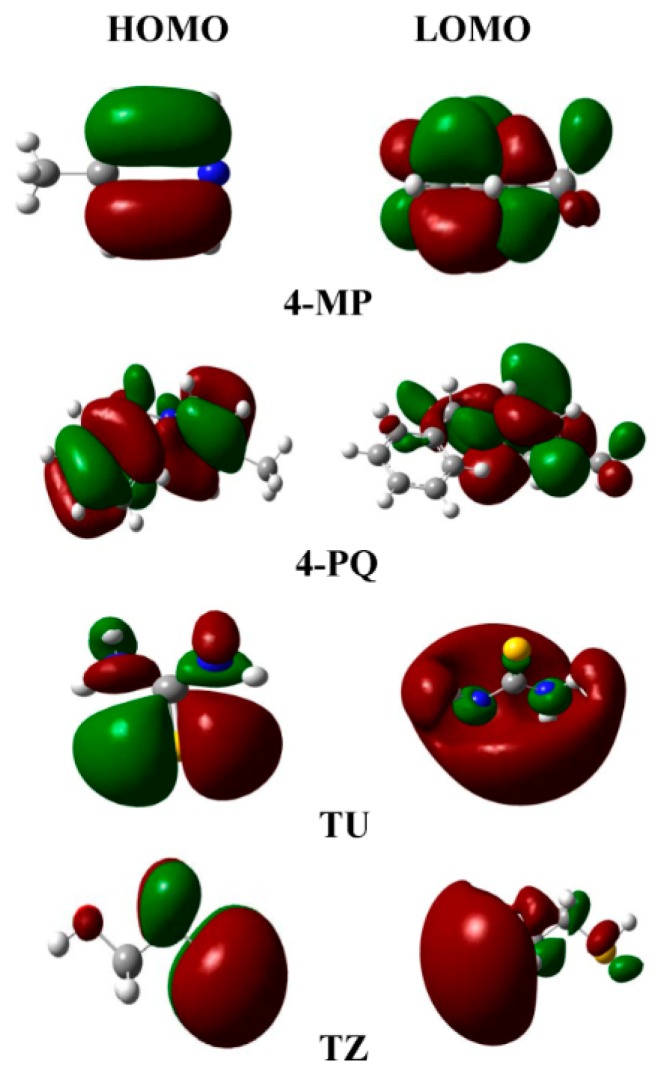
Schematic representation of the HOMO and LUMO molecular orbital of pyridine derivatives and sulfur-containing compounds (Color code: N, blue; S, yellow; C, gray; H, white).

**Figure 10 molecules-23-03270-f010:**
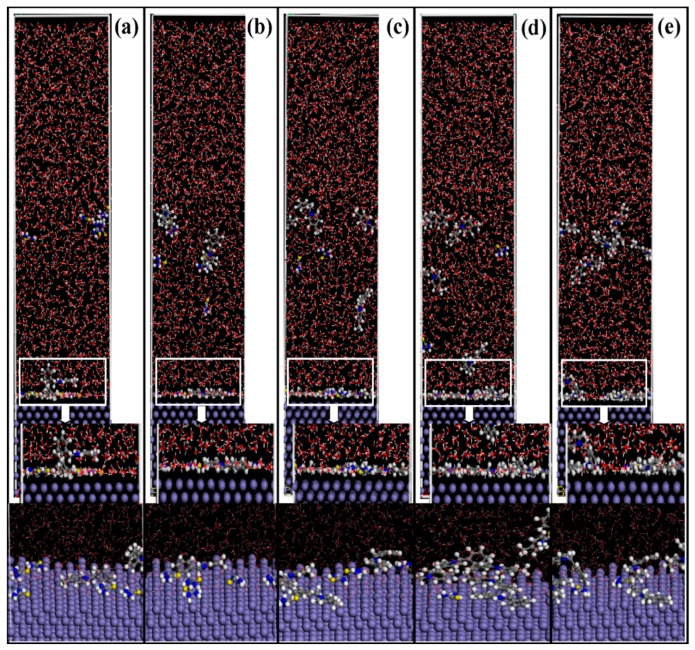
Side views of the lowest energy adsorption configurations of 4-PQ and TU with different molar ratios of (**a**) 1:5, (**b**) 1:3, (**c**) 1:1, (**d**) 3:1, and (**e**) 5:1 on Fe (1 1 0) calculated by MD simulations.

**Figure 11 molecules-23-03270-f011:**
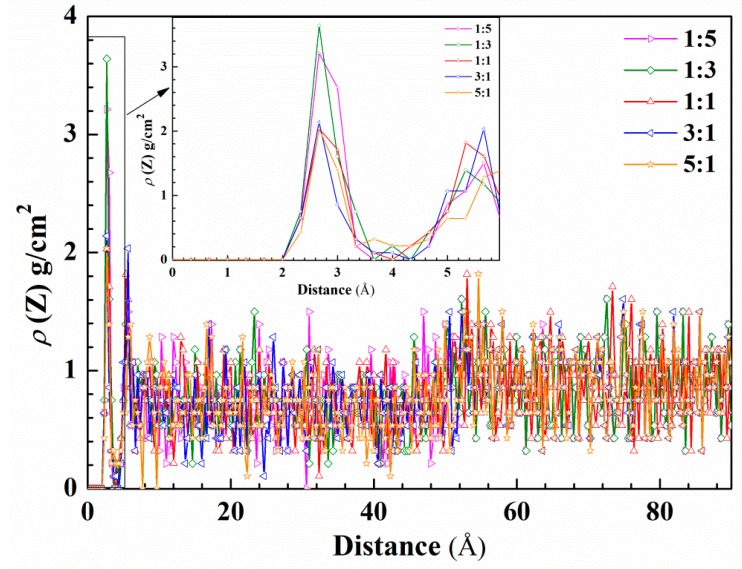
Concentration profile for water molecules on the metal surface as a function of height with corrosion inhibitors at different molar ratios.

**Figure 12 molecules-23-03270-f012:**
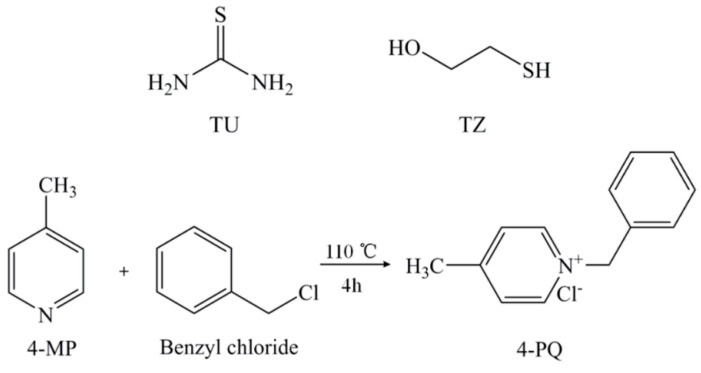
Molecular structures of TU, TZ, 4-MP, and 4-PQ, and the chemical reaction of quaternization.

**Table 1 molecules-23-03270-t001:** Corrosion rate and inhibition efficiency obtained from weight loss measurement of Q235 steel with different individual inhibitors in CO_2_-saturated 3.5 wt.% NaCl solution at 60 °C.

Inhibitors	CR (g/m^2^∙h)	IE (%)
blank	0.345 ± 0.02	-
4-MP	0.341 ± 0.01	1.4 ± 0.2
4-PQ	0.335 ± 0.02	2.7 ± 0.4
TU	0.078 ± 0.02	77.5 ± 3.1
TZ	0.076 ± 0.01	77.6 ± 2.7

**Table 2 molecules-23-03270-t002:** Polarization parameters of Q235 steel in CO_2_-saturated 3.5 wt.% NaCl solution with or without different molar ratios of 4-PQ and TZ or individual inhibitor and at 60 °C.

Inhibitors	*b*_a_ (mV)	*b*_c_ (mV)	*E*_corr_ (V)	*I*_corr_ (µA∙cm^−2^)	$\eta_{P}\;(\%)$
Blank	70 ± 3	557 ± 7	−0.726 ± 0.003	136.7 ± 10.2	
TZ	193 ± 8	379 ± 5	−0.717 ± 0.005	59.3 ± 3.3	56.6
4-PQ	80 ± 6	412 ± 6	−0.756 ± 0.002	114.6 ± 9.8	16.2
1:5	63 ± 4	173 ± 8	−0.713 ± 0.004	56.5 ± 3.2	58.6
1:3	82 ± 5	181 ± 5	−0.725 ± 0.001	90.9 ± 5.9	33.5
1:1	95 ± 4	177 ± 7	−0.710 ± 0.002	50.3 ± 4.2	63.2
3:1	60 ± 2	107 ± 5	−0.703 ± 0.002	28.4 ± 0.7	79.2
5:1	110 ± 9	165 ± 6	−0.711 ± 0.004	48.2 ± 2.5	64.7

**Table 3 molecules-23-03270-t003:** Polarization parameters of Q235 steel in CO_2_-saturated 3.5 wt.% NaCl solution with or without different molar ratios of 4-PQ and TU or individual inhibitor at 60 °C.

Inhibitors	*b*_a_ (mV)	*b*_c_ (mV)	*E*_corr_ (V)	*I*_corr_ (µA∙cm^−2^)	$\eta_{P}\;(\%)$
blank	70 ± 3	557 ± 7	−0.726 ± 0.003	136.7 ± 10.2	
TU	193 ± 8	379 ± 9	−0.714 ± 0.001	69.2 ± 7.7	49.4
4-PQ	80 ± 6	412 ± 11	−0.756 ± 0.002	114.6 ± 9.8	16.2
1:5	68 ± 5	188 ± 7	−0.710 ± 0.005	43.4 ± 3.9	68.2
1:3	67 ± 4	206 ± 8	−0.711 ± 0.004	50.2 ± 5.8	63.3
1:1	73 ± 9	172 ± 5	−0.705 ± 0.002	28.3 ± 3.6	79.2
3:1	79 ± 3	159 ± 6	−0.703 ± 0.002	20.5 ± 1.3	85.0
5:1	82 ± 7	169 ± 4	−0.701 ± 0.003	24.1 ± 2.1	82.3

**Table 4 molecules-23-03270-t004:** Equivalent circuit parameters and inhibition efficiency (IE) obtained from EIS measurements of Q235 steel in CO_2_-saturated 3.5 wt.% NaCl solution with or without different mole ratios of 4-PQ and TZ or individual inhibitor at 60 °C.

Inhibitors	*R*_s_ (Ω∙cm^2^)	*R*_f_ (Ω∙cm^2^)	*R*_ct_ (Ω∙cm^2^)	*C*_f_ (μF/cm^2^)	*n* _f_	*C*_dl_ (μF/cm^2^)	*n* _dl_	*R*_t_ (Ω∙cm^2^)	*η*_Z_ (%)	*χ* ^2^
Blank	2.58 ± 0.03	-	151.7 ± 1.51	-	-	115.9 ± 2.9	0.80 ± 0.005	151.7 ± 0.15	-	1.91 × 10^−3^
TZ	2.26 ± 0.03	0.68 ± 0.005	335.7 ± 3.36	99.8 ± 1.40	0.68 ± 0.002	73.7 ± 1.03	1 ± 0.008	336.4 ± 3.37	54.95	1.03 × 10^−4^
4-PQ	2.23 ± 0.02	0.97 ± 0.009	197.7 ± 0.20	90.9 ± 1.36	0.98 ± 0.008	85.5 ± 1.28	0.78 ± 0.004	198.7 ± 0.21	23.65	2.92 × 10^−3^
1:5	2.12 ± 0.02	1.95 ± 0.27	225.1 ± 1.58	56.0 ± 1.12	0.89 ± 0.062	74.1 ± 3.71	0.71 ± 0.004	226.1 ± 1.80	32.89	4.77 × 10^−4^
1:3	2.09 ± 0.01	0.34 ± 0.02	189.7 ± 1.70	111.7 ± 6.66	0.84 ± 0.059	94.3 ± 13.16	0.70 ± 0.004	190.5 ± 1.72	20.35	5.14 × 10^−4^
1:1	2.43 ± 0.03	1.02 ± 0.47	377.9 ± 13.23	40.3 ± 1.45	0.66 ± 0.003	67.3 ±3.30	0.75 ± 0.07	378.9 ± 13.70	59.97	9.76 × 10^−4^
3:1	2.03 ± 0.02	2.50 ± 0.27	491.1 ± 4.42	13.9 ± 0.68	0.84 ± 0.034	64.4 ± 0.26	0.68 ± 0.003	493.6 ± 4.70	69.27	5.36 × 10^−4^
5:1	2.28 ± 0.02	1.76 ± 0.04	410.0 ± 4.92	24.7 ± 1.48	0.66 ± 0.033	73.5 ± 1.18	0.93 ± 0.074	411.8 ± 4.95	63.17	2.87 × 10^−4^

**Table 5 molecules-23-03270-t005:** Equivalent circuit parameters and inhibition efficiency (IE) obtained from EIS measurements of Q235 steel in CO_2_-saturated 3.5 wt.% NaCl solution with or without different mole ratios of 4-PQ and TU or individual inhibitor at 60 °C.

Inhibitors	*R*_s_ (Ω∙cm^2^)	*R*_f_ (Ω∙cm^2^)	*R*_ct_ (Ω∙cm^2^)	*C*_f_ (μF/cm^2^)	*n* _f_	*C*_dl_ (μF/cm^2^)	*n* _dl_	*R*_t_ (Ω∙cm^2^)	*η*_Z_ (%)	*χ* ^2^
Blank	2.58 ± 0.03	-	151.7 ± 0.15	-	-	115.9 ± 2.9	0.80 ± 0.005	151.7 ± 0.15	-	1.91 × 10^−3^
TU	2.03 ± 0.02	1.54 ± 0.02	323.2 ± 4.85	21.1 ± 0.40	0.71 ± 0.004	57.0 ± 1.08	0.86 ± 0.003	324.7 ± 4.88	53.06	2.15 × 10^−4^
4-PQ	2.23 ± 0.02	0.97 ± 0.009	197.7 ± 0.20	90.9 ± 1.36	0.98 ± 0.008	85.5 ± 1.28	0.78 ± 0.004	198.7 ± 0.21	23.65	2.92 × 10^−3^
1:5	2.47 ± 0.03	1.47 ± 0.06	363.5 ± 7.26	46.3 ± 1.29	0.81 ± 0.008	89.9 ± 1.80	0.79 ± 0.031	365.0 ± 7.32	58.47	3.42 × 10^−4^
1:3	2.23 ± 0.03	2.56 ± 0.05	349.9 ± 5.24	54.9 ± 0.92	0.79 ± 0.004	98.6 ± 1.20	0.78 ± 0.010	352.5 ± 5.30	56.96	3.96 × 10^−4^
1:1	2.28 ± 0.02	1.77 ± 0.19	599.6 ± 13.78	5.7 ± 0.17	0.92 ± 0.033	56.4 ± 2.74	0.70 ± 0.009	601.4 ± 13.95	74.76	2.85 × 10^−4^
3:1	2.44 ±0.02	2.66 ± 0.48	879.6 ± 17.58	1.3 ± 0.04	0.60 ± 0.018	42.5 ± 1.68	0.72 ± 0.009	882.3 ± 18.06	82.80	3.73 × 10^−4^
5:1	2.04 ±0.01	1.66 ± 0.15	759.50 ± 15.18	18.8 ± 2.16	0.79 ± 0.010	51.2 ± 1.53	0.72 ± 0.010	761.2 ± 15.29	80.12	4.81 × 10^−4^

**Table 6 molecules-23-03270-t006:** Calculated quantum chemical indices of pyridine derivatives and sulfur-containing compounds.

Inhibitor	*E*_HOMO_ (eV)	*E*_LUMO_ (eV)	∆*E* (eV)	*η* (eV)	*ω*	∆*N*	*χ* (eV)	*μ* (kcal/mol)
4-MP	−8.648	0.069	8.717	4.358	2.111	0.311	4.289	−4.289
4-PQ	−8.123	−0.0014	8.121	4.061	2.032	0.362	4.062	−4.062
TU	−7.255	−0.110	7.146	3.573	1.898	0.464	3.682	−3.682
TZ	−8.079	0.279	8.358	4.179	1.820	0.371	3.900	−3.900

**Table 7 molecules-23-03270-t007:** The output obtained from MD simulations for the adsorption of 4-PQ and TU with different molar ratios.

4-PQ:TU	*E*_total_ (kcal/mol)	*E*_surface+solution_ (kcal/mol)	*E*_inhibitors+solution_ (kcal/mol)	*E*_solution_ (kcal/mol)	*E*_adsoption_ (kcal/mol)
1:5	−41,105.7	−39,365.3	−8953.9	−7518.03	−304.495
1:3	−40,974.3	−39,389	−8823.87	−7536.58	−297.995
1:1	−40,556.2	−39,237.3	−8347.31	−7508.61	−480.2
3:1	−39,985.6	−38,771.2	−7734.12	−7083.08	−563.43
5:1	−39,856.7	−38,861.6	−7606.54	−7157.9	−546.394

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
