# Peer review of "Experimental and Theoretical Study on the Synergistic Inhibition Effect of Pyridine Derivatives and Sulfur-Containing Compounds on the Corrosion of Carbon Steel in CO2-Saturated 3.5 wt.% NaCl Solution"

_molecules, 2018, doi:10.3390/molecules23123270_

Round 1

Reviewer 1 Report

I have no other comments.

Author Response

Thanks for your comments of this paper.

Reviewer 2 Report

Review to article:  Experimental and theoretical study on the synergistic inhibition effect of yridine derivatives and sulfur- containing compounds on the corrosion of carbon  steel in CO2-saturated 3.5 wt.% NaCl solution”

Submitted to Molecules MDPI for publications.

The article investigates effect of the different organic inhibitors on the corrosion of carbon steel in presence  CO2. In the work electrochemical methods: EIS and dc polarization Tafel extrapolation methods were used. The inhibiting efficiency was determined by gravimetric mass loss. The surface was characterized by SEM. The synergism of inhibitors, coverage of the iron surface were studied in the details.

In general the article well written, good language, good scientific level. It will be easy for readers to understand the publication.

Comments:

Introduction.

For me it is not clear, why these compounds were chosen for the study. Why you wrote about pharmacology or medical application (discussion)? This part has to be clarified in more detail. It is good to discuss the data and the structure of inhibitors, which are used in oil field industry now.    What functional groups most important, importance hydrophobic-hydrophilic properties, impact of the organic phase ( oil ) on the adsorption. These are important and interesting for corrosion points.

Line 91. Removing corrosion products. Normally is used acidic solution with addition of organic inhibitor to protect from over etching. Please clarify.

Line 134. Please clarify the initial data for modelling. Where data was taken  ELUMO and EHOMO, softness - hardness, electronegativity, etc. It have to concern some hetero-atom (S,N) or organic chain. Does it exist the reference data? Exist the real experimental data (spectroscopy e.g. XPS) that electron transfer between iron surface and N or S is really existing? Please add the reference and discussion.

Why was taken Fe (110), probably the steel surface is covered by iron oxide film. The empty orbitals on Fe can form the bonds with oxygen.

It may be useful as a model, but model has to be based on the real physical parameters and doesn’t on the some kind of calculations. At present, I don’t believe in this kind of simulation. Now it is very formal.

For example “Normal ensemble 145 (NVT) was selected, and the Andersen thermostat was used to control the temperature at 333 K.” The readers must understand you (e.g. Andersen thermostat?). I think you have to give it simpler and more understanable.

Table 2, Tafel extrapolation was used for I cor.? Please give to caption or to the text.

Line 246 Nyquist plots depressed. Can be that superposition of two semicircles (film in more high frequency and corrosion in low frequencies) give one depressed?

 Table 4 and 5, Did you simulate EIS using 2 electric schemes? Please say it in the text. The resistance of the film is very small relatively polarization resistance and probably simple Randles (RC parallel) circuit also will give good fitting. Please can you show fitting goodness for two schemes.

Line 274 “ n is the deviation parameter in regard to the phase shift.”  The phrase is not clear.

n=1 it is perfect capacitor N<1 the capacitor with leakage (leak capacitor) leakage is going through parallel resistor. In this case, capacitor is partially resistor.    

Figure 7, b. I am not sure that this circuit is valid. It normally applied to partially corroding surface (parallel two RC circuits is that some parts are non-corroding, pits or blisters in the coating). You have uniform corrosion in all cases. May be better in this case use consecutive connection of two parallel RC circuits.

Line 298 “the linear polarization resistance method was used”. It is special method when the potential scans around OCP for +-20 mV. You didn’t say anything about this method in Experimental. You use Tafel extrapolation of dc polarization curves Figure 8,b (right?).   

Table 6, what are the parameters, explain in caption.

Line 373, why the iron and not the iron oxide?

Figure 11. No any molecules of water (white balls?) at the interface in all cases? Contradictts with Figure 12 there was calculated water at the distance 2.5 Å.

Table 7, The units of the energies?

I think you have to note that it is just a modelling, it can be far from reality. Is it give something to choose more effective inhibitor?

Author Response

Response to Reviewer 2 Comments

Point 1: Introduction. For me it is not clear, why these compounds were chosen for the study. Why you wrote about pharmacology or medical application (discussion)? This part has to be clarified in more detail. It is good to discuss the data and the structure of inhibitors, which are used in oil field industry now. What functional groups most important, importance hydrophobic-hydrophilic properties, impact of the organic phase (oil) on the adsorption. These are important and interesting for corrosion points.

Response 1: Thanks for your valuable comments. Introduction has been adjusted and modified in the manuscript. The discussion of commonly used corrosion inhibitors in oilfield (structure and performance), and the reasons of selecting pyridine derivatives and sulfur-containing compounds to study have been added. The experimental and theoretical investigation methods in references and the significance of probing synergistic mechanism were also discussed in the revised version.

Point 2: Line 91. Removing corrosion products. Normally is used acidic solution with addition of organic inhibitor to protect from over etching. Please clarify.

Response 2: Thanks for your suggestion. The sentence has been modified as follows:immersed in the acid bath solution with hexamethylenetetramine for a short time to remove the corrosion product and protect from over etching”.

Point 3: Line 134. Please clarify the initial data for modelling. Where data was taken ELUMO and EHOMO, softness - hardness, electronegativity, etc. It has to concern some hetero-atom (S, N) or organic chain. Does it exist the reference data? Exist the real experimental data (spectroscopy e.g. XPS) that electron transfer between iron surface and N or S is really existing? Please add the reference and discussion.

Response 3: Sorry for the unclear information. The method of Density Functional Theory (DFT) M06-2x/6-311+G (d, p) was used to optimized initial structure of the corrosion inhibitor molecules, due to the corrosion inhibitor molecules are composed of main group elements [1,2]. And the following parameters can be directly obtained. The Fukui function was used to analyse the local active sites of the corrosion inhibitors and predict the way of molecules adsorption on the metal surface [3].

In a number of experimental studies, the adsorption of organic inhibitors relies on the lone pair electrons to form covalent bonds with the orbitals of metal atoms. Fe-N adsorption bond (B.E. 399.6±0.1eV) can be formed by the adsorption of the pyridine, quinoline and the ammonium groups of quaternary ammonium salt on the steel surface [4,5], and the bonding between Fe and C-N-H2 (B.E. 400.9±0.3eV for N1s) from thiourea [6,7] also was detected by XPS. The XPS spectrum around B.E. 162-164  eV, which is often observed on the steel surface when use S-containing corrosion inhibitors, can be ascribed to Fe-S adsorption bond [8-10].

Quantum chemical studies are one of the important tools, which act as a bridge between experimental and theoretical studies. DFT was used for predicting the geometric and electronic properties of corrosion inhibitors. However, the calculated parameters are not completely consistent with the experimental data, they are studied for individual molecule and these parameters reflect energy tendencies of the molecules to explain the same change. In general, a high value of EHOMO represents a higher electron donating ability of the inhibitor and therefore high bonding tendency between the metal and the inhibitor. Similarly, a lower value of ELUMO represents the higher electron accepting tendency of the inhibitor from metal surface. Therefore it can be concluded that the higher value of EHOMO and lower value of ELUMO facilitate the inhibitor bonding between metal and inhibitor. Similarly, a lower value of ΔE is associated with a high chemical reactivity and therefore high inhibition efficiency. Moreover, lower electronegativity (χ), higher value of fraction of electron transfer (ΔN) represent the greater electron donating ability of the inhibitor. While, higher value of absolute chemical hardness (η) and electronegativity decreases the electron donating ability of the inhibitor molecule, etc [5,11].

Molecular dynamics (MD) simulation, which has been proven to be very efficient in the evaluation of the interaction of organic compounds with metal surface. It takes into account the interaction between the inhibitor molecules and the metal surface (adsorption sites or oxide sites between the N, π- electron or S and iron surface) [12], the inhibitor molecules and the corrosive medium and the different molecules (intermolecular force and competitive adsorption) [13–15]. In this paper, the molecular dynamics simulation of the corrosion inhibitors mixture considering the H2O molecule, kinds of corrosion inhibitor composition and concentration has been proposed to close the reality. The modelling can be successfully used as reliable approaches to coincide with experimental results and explain the synergistic inhibition mechanism.

Point 4: Why was taken Fe (110), probably the steel surface is covered by iron oxide film. The empty orbitals on Fe can form the bonds with oxygen.

Response 4: In fact, corrosion inhibitors do adsorb on the iron oxides and metallic Fe surfaces at the same time. However, for carbon steel in oilfield environment (oxygen depleted), the corrosion mechanism is cathodic hydrogen reduction. Its oxide film coverage is often incomplete or loose, and the protection mechanism of organic corrosion inhibitor is widely accepted as the barrier effect by surface adsorption of molecule on metallic iron surface.

The Fe (110) surface can represent the typical state of the surface [12]. Because, firstly Fe (111) and Fe (100) surfaces have relatively open structures compared with the other three kinds of Fe surfaces (110, 100, 111), secondly Fe (110) is a density packed surface and has the most stabilization [15,16]. It is difficult to design an reliable iron metal and iron oxide combined surface in molecular dynamic model at present due to the limited scale of interface (to save the calculation time). So we choose Fe (110) surface to simulate the adsorption process.

It is worth to study the adsorption of corrosion inhibitors on the complicated surface considering the existing of oxide film within metallic iron, and we will consider this situation in our future work. Thank you very much for this valuable comment.

Point 5: It may be useful as a model, but model has to be based on the real physical parameters and doesn’t on the some kind of calculations. At present, I don’t believe in this kind of simulation. Now it is very formal.

For example “Normal ensemble 145 (NVT) was selected, and the Andersen thermostat was used to control the temperature at 333 K.” The readers must understand you (e.g. Andersen thermostat?). I think you have to give it simpler and more understandable.

Response 5: We totally agree with the reviewer’s comment that the model has to be based on the real physical parameters. For the synergistic mechanism of mixed corrosion inhibitors, the hypothesis is proposed through the experimental results firstly in this paper. The theoretical calculation is performed to verify the experimental results based on the complex real conditions.

Especially for the molecular dynamics (MD) simulation, which has been proven to be very efficient in the evaluation of the interaction of organic compounds with metal surface. It takes into account the interaction between the inhibitor molecules and the metal surface [12], the inhibitor molecules and the corrosive medium and the different molecules (intermolecular force and competitive adsorption) [13–15]. In this paper, the molecular dynamics simulation of the corrosion inhibitors mixture considering the H2O molecule is conducted. The corrosion inhibitor composition and concentration have been set to close the reality, and the temperature has been set to close the experimental conditions. The modelling can be successfully used as reliable approaches to coincide with experimental results and explain the synergistic inhibition mechanism.

The description in the method description has been revised to give more details. However, because of the computer calculation speed limitation, the present simulation model still is a simplified model based on complex experimental conditions. We will develop the simulation model more and more close to real condition with the development of computer calculation capacity.

Point 6: Table 2, Tafel extrapolation was used for I cor.? Please give to caption or to the text.

Response 6: Thanks for your suggestion. The sentence has been supplemented as follows: “The fitting results of the polarization curves used Tafel extrapolation by software C-view, in which other electrochemical parameters and the inhibition efficiencies of inhibitors can be determined from the corrosion current densities (Icorr) are shown in Table 2 and 3, respectively”.

Point 7: Line 246 Nyquist plots depressed. Can be that superposition of two semicircles (film in more high frequency and corrosion in low frequencies) give one depressed?

Response 7: The Nyquist plots obtained for the corrosion of Q235 in CO2 containing solutions with inhibitors consist of a depressed capacitive loop. It can be considered that the resistance of inhibitors film is much smaller than the charge transfer resistance. The semicircle representing the inhibitor film merging with the charge transfer loop result in that the Nyquist plots emerge a “degradation” phenomenon [18]. In the Bode plots, it can be seen that there are two-time constants. One time constant emerges at a frequency about 10 Hz, this can be attributed to the double layer capacitance and charge transfer resistance. Another at about 1000 Hz can be corresponding to the relaxation process of an adsorbed inhibitor film as a surface dielectric film normally has a small time constant and so has a phase angle shift in the high frequency range. The inhibitors film changes the electrode interfacial structure and results in an additional time constant.

Point 8: Table 4 and 5, Did you simulate EIS using 2 electric schemes? Please say it in the text. The resistance of the film is very small relatively polarization resistance and probably simple Randles (RC parallel) circuit also will give good fitting. Please can you show fitting goodness for two schemes.

Response 8: Thanks for your valuable comments. Yes, we have simulated EIS using 2 electric schemes. The EIS plots of Figure 1 shows fitting curves by R(CR) and experimental points of different mole ratios of 4-PQ and TU. Table 1 and 2 show the parameters of different mole ratios of 4-PQ and TU fitted by equivalent circuits of R(CR) and R(Q(R(QR))). Overall, if a simple circuit (RC parallel) is used to fit, the fitting error is large, the goodness of fit (χ2) is always more than 10−2. However, the fitting results match the experimental data very well while used the equivalent circuit of Figure 7 in manuscript, and the χ2 is always less than10−3.

Figure. 1. Nyquist of R(CR) fitted for different molar ratios of 4-PQ and TU.

Table 1. Equivalent circuit parameters by R(CR) fitting with different mole ratios of 4-PQ and TU.

Inhibitors

Rs

(Ω∙cm2)

Rct

(Ω∙cm2)

Cdl

(μF/ cm2)

χ2

1:5

3.26

311.5

614.3

7.85E-2

1:3

4.44

316.8

466.7

7.64E-2

1:1

3.30

486.8

199.2

1.33E-1

3:1

3.56

714.2

124.7

1.49E-1

5:1

3.58

610.9

161.8

1.68E-1

Table 2. Equivalent circuit parameters by R(Q(R(QR))) fitting with different mole ratios of 4-PQ and TU.

Inhibitors

Rs

(Ω∙cm2)

Rf

(Ω∙cm2)

Rct

(Ω∙cm2)

Cf

(μF/ cm2)

nf

Cdl

(μF/ cm2)

ndl

χ2

1:5

2.47

1.47

363.5

46.3

0.81

89.9

0.79

3.42E-4

1:3

2.23

2.56

349.9

54.9

0.79

98.6

0.78

3.96E-4

1:1

2.28

1.77

599.6

5.7

0.92

56.4

0.70

2.85E-4

3:1

2.44

2.66

879.6

1.3

0.60

42.5

0.72

3.73E-4

5:1

2.04

1.66

759.50

18.8

0.79

51.2

0.72

4.81E-4

When modelling EIS plots in corrosion investigation, a pure double-layer capacitor is frequently replaced by Constant Phase Elements (CPE) representing leaky or non-ideal capacitors with a view to compensating for non-homogeneity in the system in order to give a more accurate fit. The impedance of CPE is expressed as the following:

where Y0 is the constant, j is the imaginary number (j2 = -1), w is the angular frequency, and value of n represents the deviation from the ideal behavior and it lies between 0 and 1. From the fitting results, the value of n is between 0.7 and 0.9.

The values of the two CPE exponents (Cdl and Cf) presented in the table support our opinion that CPEdl and CPEf can be prescribed to the capacitive behaviour of the electrochemical double-layer and the adsorbed inhibitors layer, respectively [19]. Further, the generally decreasing trend of Cf with changing molar ratios about 4-PQ: TZ or TU in the solution is as expected (3:1 < 5:1 < 1:1 < 1:5 < 1:3), and indicates the formation of inhibitors film on the metal surface. The trend in CPEdl behaviour is also logical, i.e. as the surface coverage of inhibitors film increases, the corresponding CPEdl value decreases.

Point 9: Line 274 “n is the deviation parameter in regard to the phase shift.”  The phrase is not clear. n=1 it is perfect capacitor N<1 the capacitor with leakage (leak capacitor) leakage is going through parallel resistor. In this case, capacitor is partially resistor.   

Response 9: Thanks for your suggestion. The sentence has been modified as follows: “The value of n represents the deviation from the ideal behavior and it lies between 0 and 1”.

Point 10: Figure 7, b. I am not sure that this circuit is valid. It normally applied to partially corroding surface (parallel two RC circuits is that some parts are non-corroding, pits or blisters in the coating). You have uniform corrosion in all cases. May be better in this case use consecutive connection of two parallel RC circuits.

Response 10: At the beginning, we have tried to use one RC circuit and two RC circuits to fit, but it can not apply to most mixed corrosion inhibitors in this investigation because the errors are big. According to references [20,21], the inhibition performance of the mixed corrosion inhibitors may be based on the density of adsorption film on the metal surface but without solid experimental or theoretical evidences. In this investigation, the hypothesis has been confirmed by the molecular dynamics simulation results theoretically to a certain extent. It can be clearly seen from “Figure 11” in the manuscript that the mixed inhibitors molecules directly covered the surface of iron to form a single inhibition film. Therefore, the circuit proposed in the paper was used for fitting according to the physical model of interface, and all errors were less than 10-3. The equivalent adsorption model is shown in Figure 2. The adsorption film density relies on inhibitor types and the molar ratio of mixed corrosion inhibitors. Also the adsorption and desorption of meceis completing during all the

Figure. 2. The equivalent adsorption model

Point 11: Line 298 “the linear polarization resistance method was used”. It is special method when the potential scans around OCP for +-20 mV. You didn’t say anything about this method in Experimental. You use Tafel extrapolation of dc polarization curves Figure 8, b (right?).  

Response 11: Sorry for the unclear description. The paper describes the method in the last paragraph of 2.3. Electrochemical measurement experiments, and has been revised as follows: To study the order of adsorption in synergistic effect, the liner polarization resistance tests were performed for different sequence of adding the one inhibitor at intervals of 40 minutes under a potential range from –10 mV to +10 mV (vs. OCP) at sweep rate of 0.2 mV s-1 at 10 minutes interval.”. Figure 3 presents the curves of linear polarization and Table 3 shows the supplementary data of the polarization resistances were obtained using linear fitting by software C-view.

Figure. 3. The curve of linear polarization under a potential range from –10 mV to +10 mV (vs. OCP) at sweep rate of 0.2 mV s-1 at 10 minutes interval: (a) blank, (b) adding mixture inhibitors contained 4-PQ and TU at the beginning, (c) adding 4-PQ at beginning, then adding TU at the time of 40th minutes, (d) adding TU at beginning, then adding 4-PQ at the time of 40th minutes.

Table 3. The polarization resistance calculated by linear polarization curve fitting.

Time   (min)

Rp (Ω∙cm2)

(a)

(b)

(c)

(d)

5

135.58

757.6

423.47

215.16

10

131.64

763.9

455.00

255.58

15

139.47

745.06

453.09

279.79

20

151.50

742.52

458.41

280.56

25

148.29

748.56

564.69

494.49

30

157.31

773.79

610.34

561.52

35

152.08

777.88

695.44

585.31

40

155.76

785.90

740.66

610.62

Point 12: Table 6, what are the parameters, explain in caption.

Response 12: Very sorry for our carelessness. The parameters symbols may be changed due to format modification during submitting. We have revised Table 6. The parameters in Table 6 are also explained in 2.5. Computational details as follows: The relevant quantum chemical parameters including energies of LUMO (ELUMO) and HOMO (EHOMO), energy gap (ΔE), absolute chemical hardness (η), electrochemical potential (μ), electronegativity (χ), the number of electron transfer (ΔN), electrophilicity index (ω) for four inhibitors were considered.”, and added the reference in the corresponding position of the paper. In addition, a supplementary explanation was added on line 398.

Point 13: Line 373, why the iron and not the iron oxide?

Response 13: As the explanation in Point 4. This paper mainly discussed the adsorption mechanism on iron. We will further consider the influence of iron oxide in following work in related investigation. Thanks very much for your suggestion.

Point 14: Figure 11. No any molecules of water (white balls?) at the interface in all cases? Contradicts with Figure 12 there was calculated water at the distance 2.5 Å.

Response 14: In Figure 11, five models containing 12 corrosion inhibitor molecules and 1000 water molecules are constructed. The water molecules look like red mesh, which is set to stick model to highlight the corrosion inhibitor molecules composed of N (blue ball), S (yellow ball), C (gray ball) and H (white ball). It can be observed from Figure 11 that mixed inhibitors of different molar ratios pressed out water molecules on the surface of the iron. This is consistent with the results shown in Figure 12. The distance of molecules began to adsorb on the Fe (1 1 0) surface is about 2.5 Å (<2 Å is the vacuum layer). And because of mixed inhibitors of different molar ratios adsorption, the density of water molecules varies with distance.

Point 15: Table 7, The units of the energies?

Response 15: Very sorry for our carelessness. The unit of energies is “kcal/mol” and has been revised in Table 7.

Point 16: I think you have to note that it is just a modelling, it can be far from reality. Is it give something to choose more effective inhibitor?

Response 16: Thanks for your valuable comment. Indeed, the real adsorption configuration at the metal/solution interface is very difficult to be confirmed by the in-situ microscopic characterization in molecule-level. In this paper, the modelling for molecule dynamic simulations respect to the actual conditions (in aqueous solution, molecular concentration, different ratios of the two molecules), and provides a new method to explain the way of mixed inhibitors adsorption and better to combine with experiment results.

However, we totally understand present modelling may be still far from the reality. We have to develop the experimental approaches along with the modelling methods to understand this nonequilibrium adsorption-desorption process at interface. Thanks very much for your opinion.

It is significant to use the similar conditions on the basis of the current model for choosing more effective inhibitor in our works. It is one of our aims. For instance, using this model, the possible optimal molar ratios can be selected for achieving better inhibition synergistic effect of the two mixed corrosion inhibitors before the experimental evaluation. It can be applied for selecting mixed corrosion inhibitor formulations, too.

Reference

1.       Yilmaz, N.; Fitoz, A.; Ergun, Ü.; Emregül, K.C. A combined electrochemical and theoretical study into the effect of 2-((thiazole-2-ylimino)methyl)phenol as a corrosion inhibitor for mild steel in a highly acidic environment. Corros. Sci. 2016, 111, 110–120, doi:10.1016/j.corsci.2016.05.002.

2.       Ansari, K.R.; Quraishi, M.A.; Singh, A. Pyridine derivatives as corrosion inhibitors for N80 steel in 15% HCl: Electrochemical, surface and quantum chemical studies. Meas. J. Int. Meas. Confed. 2015, doi:10.1016/j.measurement.2015.08.028.

3.       Obot, I.B.; Macdonald, D.D.; Gasem, Z.M. Density functional theory (DFT) as a powerful tool for designing new organic corrosion inhibitors: Part 1: An overview. Corros. Sci. 2015.

4.       Zhao, J.; Duan, H.; Jiang, R. Synergistic corrosion inhibition effect of quinoline quaternary ammonium salt and Gemini surfactant in H2S and CO2saturated brine solution. Corros. Sci. 2015, doi:10.1016/j.corsci.2014.11.007.

5.       Hu, K.; Zhuang, J.; Ding, J.; Ma, Z.; Wang, F.; Zeng, X. Influence of biomacromolecule DNA corrosion inhibitor on carbon steel. Corros. Sci. 2017, doi:10.1016/j.corsci.2017.06.004.

6.       Nelson, G.W.; Perry, M.; He, S.M.; Zechel, D.L.; Horton, J.H. Characterization of covalently bonded proteins on poly(methyl methacrylate) by X-ray photoelectron spectroscopy. Colloids Surfaces B Biointerfaces 2010, doi:10.1016/j.colsurfb.2010.02.012.

7.       Fertier, L.; Rolland, M.; Thami, T.; Persin, M.; Zimmermann, C.; Lachaud, J.L.; Rebière, D.; Déjous, C.; Bêche, E.; Cretin, M. Synthesis and grafting of a thiourea-based chelating agent on SH-SAW transducers for the preparation of thin films sensitive to heavy metals. Mater. Sci. Eng. C 2009, doi:10.1016/j.msec.2008.07.032.

8.       Zhang, C.; Duan, H.; Zhao, J. Synergistic inhibition effect of imidazoline derivative and L-cysteine on carbon steel corrosion in a CO2-saturated brine solution. Corros. Sci. 2016, doi:10.1016/j.corsci.2016.07.018.

9.       Kashkovskiy, R. V.; Kuznetsov, Y.I.; Kazansky, L.P. Inhibition of hydrogen sulfide corrosion of steel in gas phase by tributylamine. Corros. Sci. 2012, 64, 126–136, doi:10.1016/j.corsci.2012.07.010.

10.     Mullet, M.; Boursiquot, S.; Abdelmoula, M.; Génin, J.M.; Ehrhardt, J.J. Surface chemistry and structural properties of mackinawite prepared by reaction of sulfide ions with metallic iron. Geochim. Cosmochim. Acta 2002, doi:10.1016/S0016-7037(01)00805-5.

11.     Kosari, A.; Moayed, M.H.; Davoodi, A.; Parvizi, R.; Momeni, M.; Eshghi, H.; Moradi, H. Electrochemical and quantum chemical assessment of two organic compounds from pyridine derivatives as corrosion inhibitors for mild steel in HCl solution under stagnant condition and hydrodynamic flow. Corros. Sci. 2014, 78, 138–150, doi:10.1016/j.corsci.2013.09.009.

12.     Zhang, Z.; Tian, N.C.; Huang, X.D.; Shang, W.; Wu, L. Synergistic inhibition of carbon steel corrosion in 0.5 M HCl solution by indigo carmine and some cationic organic compounds: Experimental and theoretical studies. RSC Adv. 2016, 6, 22250–22268, doi:10.1039/c5ra25359d.

13.     Xia, S.; Qiu, M.; Yu, L.; Liu, F.; Zhao, H. Molecular dynamics and density functional theory study on relationship between structure of imidazoline derivatives and inhibition performance. Corros. Sci. 2008, 50, 2021–2029, doi:10.1016/j.corsci.2008.04.021.

14.     Amer, R.; El-Sherif, A.A.; Ebrahim, H.; Mokhtar, A. Stability analysis for multi-user cooperative cognitive radio network with energy harvesting. 2016 2nd IEEE Int. Conf. Comput. Commun. ICCC 2016 - Proc. 2017, 10, 2369–2375, doi:10.1016/j.surfin.2017.11.007.

15.     Khaled, K.F. Molecular modeling and electrochemical investigations of the corrosion inhibition of nickel using some thiosemicarbazone derivatives. J. Appl. Electrochem. 2011, 41, 423–433, doi:10.1007/s10800-010-0252-1.

16.     El-Taib Heakal, F.; Rizk, S.A.; Elkholy, A.E. Characterization of newly synthesized pyrimidine derivatives for corrosion inhibition as inferred from computational chemical analysis. J. Mol. Struct. 2018, 1152, 328–336, doi:10.1016/j.molstruc.2017.09.079.

17.     Satoh, S.; Fujimoto, H.; Kobayashi, H. Theoretical study of NH3 adsorption on Fe(110) and Fe(111) surfaces. J. Phys. Chem. B 2006, doi:10.1021/jp055097w.

18.     Zhang, G.; Chen, C.; Lu, M.; Chai, C.; Wu, Y. Evaluation of inhibition efficiency of an imidazoline derivative in CO2-containing aqueous solution. Mater. Chem. Phys. 2007, doi:10.1016/j.matchemphys.2007.04.076.

19.     Georges, C.; Rocca, E.; Steinmetz, P. Synergistic effect of tolutriazol and sodium carboxylates on zinc corrosion in atmospheric conditions. Electrochim. Acta 2008, 53, 4839–4845, doi:10.1016/j.electacta.2008.01.073.

20.     Tian, H.; Li, W.; Hou, B.; Wang, D. Insights into corrosion inhibition behavior of multi-active compounds for X65 pipeline steel in acidic oilfield formation water. Corros. Sci. 2017, doi:10.1016/j.corsci.2017.01.010.

21.     Desimone, M.P.; Grundmeier, G.; Gordillo, G.; Simison, S.N. Amphiphilic amido-amine as an effective corrosion inhibitor for mild steel exposed to CO2saturated solution: Polarization, EIS and PM-IRRAS studies. Electrochim. Acta 2011, 56, 2990–2998, doi:10.1016/j.cej.2018.05.059.

Reviewer 3 Report

The manuscript is within the scope of Molecules. The manuscript is well organized nevertheless the English needs to be improved. Before publication some minor concerns must be addressed:

1 - The introduction needs to be updated since does not reflect the actual state of the art.

2 - Impedance data in Figures 6 a) and b) are not plotted correctly. Nyquist plots should have equal or square axes. That is dimensions of the same magnitude along each axis.

3 - Table 1 - Please introduce the errors.

4 - Tables 2 and 3 - Please introduce the errors obtained for the Icorr.

4 - Tables 4 and 5 - Please introduce the errors and the goodness of the fitting (chi^2)

5 - Electrochemical techniques - Please introduce how many samples were studied in each measurement.

Author Response

Response to Reviewer 3 Comments

Point 1: The introduction needs to be updated since does not reflect the actual state of the art.

Response 1: Thanks for your valuable comments. Introduction has been adjusted and revised in the manuscript.

Point 2: Impedance data in Figures 6 a) and b) are not plotted correctly. Nyquist plots should have equal or square axes. That is dimensions of the same magnitude along each axis.

Response 2: Thanks for your suggestion. Figure 6(a) and (b) have been revised in manuscript as follows:

Figure. 1. Nyquist (a) and (b) of Q235 steel in CO2-saturated 3.5 wt.% NaCl solution with or without different mole ratios of 4-PQ and sulfur-containing compounds or individual inhibitor at 60°C.

Point 3: Table 1 - Please introduce the errors.

Response 3: Thanks for your suggestions. Before and after each weight loss experiment, the coupons were dried and weighed using an analytical balance (precision ± 0.1 mg) and the mean weight loss value is reported and the corresponding standard deviation has been added in the revised manuscript as follows:

Table 1. Corrosion rate and inhibition efficiency obtained from weight loss measurement of Q235 steel with different individual inhibitor in CO2-saturated 3.5 wt.% NaCl solution at 60°C.

Inhibitors

CR (g/m2∙h)

IE (%)

blank

0.345 ± 0.02

-

4-MP

0.341 ± 0.01

1.4 ± 0.2

4-PQ

0.335 ± 0.02

2.7 ± 0.4

TU

0.078 ± 0.02

77.5 ± 3.1

TZ

0.076 ± 0.01

77.6 ± 2.7

Point 4: Tables 2 and 3 - Please introduce the errors obtained for the Icorr.

Response 4: Thanks for your suggestions. The related result has been added in the Tables 2 and 3 as follows: Each electrochemical measurement was repeated four times under the same conditions to ensure reliability.

Table 2. Polarization parameters of Q235 steel in CO2-saturated 3.5 wt.% NaCl solution with or without different mole ratios of 4-PQ and TZ or individual inhibitor and at 60°C.

Inhibitors

ba (mV)

-bc (mV)

Ecorr(V)

Icorr(µA∙cm-2)

ηp (%)

Blank

70 ± 3

557 ± 7

-0.726 ± 0.003

136.7 ± 10.2

TZ

193 ± 8

379 ± 5

-0.717 ± 0.005

59.3 ± 3.3

56.6

4-PQ

80 ± 6

412 ± 6

-0.756 ± 0.002

114.6 ± 9.8

16.2

15

63 ± 4

173 ± 8

-0.713 ± 0.004

56.5 ± 3.2

58.6

13

82 ± 5

181 ± 5

-0.725 ± 0.001

90.9 ± 5.9

33.5

11

95 ± 4

177 ± 7

-0.710 ± 0.002

50.3 ± 4.2

63.2

31

60 ± 2

107 ± 5

-0.703 ± 0.002

28.4 ± 0.7

79.2

51

110 ± 9

165 ± 6

-0.711 ± 0.004

48.2 ± 2.5

64.7

Table 3. Polarization parameters of Q235 steel in CO2-saturated 3.5 wt.% NaCl solution with or without different mole ratios of 4-PQ and TU or individual inhibitor at 60°C.

Inhibitors

ba (mV)

-bc (mV)

Ecorr(V)

Icorr(µA∙cm-2)

ηp (%)

blank

70 ± 3

557 ± 7

-0.726 ± 0.003

136.7 ± 10.2

TU

193 ± 8

379 ± 9

-0.714 ± 0.001

69.2 ± 7.7

49.4

4-PQ

80 ± 6

412 ± 11

-0.756 ± 0.002

114.6 ± 9.8

16.2

15

68 ± 5

188 ± 7

-0.710 ± 0.005

43.4 ± 3.9

68.2

13

67 ± 4

206 ± 8

-0.711 ± 0.004

50.2 ± 5.8

63.3

11

73 ± 9

172 ± 5

-0.705 ± 0.002

28.3 ± 3.6

79.2

31

79 ± 3

159 ± 6

-0.703 ± 0.002

20.5 ± 1.3

85.0

51

82 ± 7

169 ± 4

-0.701 ± 0.003

24.1 ± 2.1

82.3

Point 5: Tables 4 and 5 - Please introduce the errors and the goodness of the fitting (chi^2)

Response 5: Thanks for your suggestions. The related result and discussion have been added in the revised manuscript as follows:

Table 4. Equivalent circuit parameters and inhibition efficiency (IE) obtained from EIS measurements of Q235 steel in CO2-saturated 3.5 wt.% NaCl solution with or without different mole ratios of 4-PQ and TZ or individual inhibitor at 60°C.

Inhibitors

Rs

(Ω∙cm2)

Rf

(Ω∙cm2)

Rct

(Ω∙cm2)

Cf

(μF/cm2)

nf

Cdl

(μF/ cm2)

ndl

Rt

(Ω∙cm2)

ηz

(%)

χ2

Blank

2.58 ± 0.03

-

151.7 ± 1.51

-

-

115.9 ± 2.9

0.80 ± 0.005

151.7 ± 0.15

-

1.91E-3

TZ

2.26 ± 0.03

0.68 ± 0.005

335.7 ± 3.36

99.8 ± 1.40

0.68 ± 0.002

73.7 ± 1.03

1 ± 0.008

336.4 ± 3.37

54.95

1.03E-4

4-PQ

2.23 ± 0.02

0.97 ± 0.009

197.7 ± 0.20

90.9 ± 1.36

0.98 ± 0.008

85.5 ± 1.28

0.78 ± 0.004

198.7 ± 0.21

23.65

2.92E-3

1:5

2.12 ± 0.02

1.95 ± 0.27

225.1 ± 1.58

56.0 ± 1.12

0.89 ± 0.062

74.1 ± 3.71

0.71 ± 0.004

226.1 ± 1.80

32.89

4.77E-4

1:3

2.09 ± 0.01

0.34 ± 0.02

189.7 ± 1.70

111.7 ± 6.66

0.84 ± 0.059

94.3 ± 13.16

0.70 ± 0.004

190.5 ± 1.72

20.35

5.14E-4

1:1

2.43 ± 0.03

1.02 ± 0.47

377.9 ± 13.23

40.3 ± 1.45

0.66 ± 0.003

67.3 ±3.30

0.75 ± 0.07

378.9 ± 13.70

59.97

9.76E-4

3:1

2.03 ± 0.02

2.50 ± 0.27

491.1 ± 4.42

13.9 ± 0.68

0.84 ± 0.034

64.4 ± 0.26

0.68 ± 0.003

493.6 ± 4.70

69.27

5.36E-4

5:1

2.28 ± 0.02

1.76 ± 0.04

410.0 ± 4.92

24.7 ± 1.48

0.66 ± 0.033

73.5 ± 1.18

0.93 ± 0.074

411.8 ± 4.95

63.17

2.87E-4

Table 5. Equivalent circuit parameters and inhibition efficiency (IE) obtained from EIS measurements of Q235 steel in CO2-saturated 3.5 wt.% NaCl solution with or without different mole ratios of 4-PQ and TU or individual inhibitor at 60°C.

Inhibitors

Rs

(Ω∙cm2)

Rf

(Ω∙cm2)

Rct

(Ω∙cm2)

Cf

(μF/ cm2)

nf

Cdl

(μF/ cm2)

ndl

Rt

(Ω∙cm2)

ηz

(%)

χ2

Blank

2.58 ± 0.03

-

151.7 ± 0.15

-

-

115.9 ± 2.9

0.80 ± 0.005

151.7 ± 0.15

-

1.91E-3

TU

2.03 ± 0.02

1.54 ± 0.02

323.2 ± 4.85

21.1 ± 0.40

0.71 ±

57.0 ± 1.08

0.86 ±

324.7 ± 4.88

53.06

2.15E-4

4-PQ

2.23 ± 0.02

0.97 ± 0.009

197.7 ± 0.20

90.9 ± 1.36

0.98 ± 0.008

85.5 ± 1.28

0.78 ± 0.004

198. 7 ± 0.21

23.65

2.92E-3

1:5

2.47 ± 0.03

1.47 ± 0.06

363.5 ± 7.26

46.3 ± 1.29

0.81 ± 0.008

89.9 ± 1.80

0.79 ± 0.031

365.0 ± 7.32

58.47

3.42E-4

1:3

2.23 ± 0.03

2.56 ± 0.05

349.9 ± 5.24

54.9 ± 0.92

0.79 ± 0.004

98.6 ± 1.20

0.78 ± 0.010

352.5 ± 5.30

56.96

3.96E-4

1:1

2.28 ± 0.02

1.77 ± 0.19

599.6 ± 13.78

5.7 ± 0.17

0.92 ± 0.033

56.4 ± 2.74

0.70 ± 0.009

601.4 ± 13.95

74.76

2.85E-4

3:1

2.44 ±0.02

2.66 ± 0.48

879.6 ± 17.58

1.3 ± 0.04

0.60 ± 0.018

42.5 ± 1.68

0.72 ± 0.009

882.3 ± 18.06

82.80

3.73E-4

5:1

2.04 ±0.01

1.66 ± 0.15

759.50 ± 15.18

18.8 ± 2.16

0.79 ± 0.010

51.2 ± 1.53

0.72 ± 0.010

761.2 ± 15.29

80.12

4.81E-4

Point 6: Electrochemical techniques - Please introduce how many samples were studied in each measurement.

Response 6: Thanks for your valuable questions. The solution and working electrode were changed after each sweep. Three to four measurements were performed for each experimental condition to estimate the repeatability, all the values reported in the paper represent mean values of at least three replicate experiments. Thanks very much for your suggestions.

Round 2

Reviewer 3 Report

All the issues raised were correctly addressed. I suggest english language revision.